# Subclone-specific microenvironmental impact and drug response in refractory multiple myeloma revealed by single-cell transcriptomics

Stephan M. Tirier[1], Jan-Philipp Mallm[1,2,3], Simon Steiger [1], Alexandra M. Poos[4,5], Mohamed H. S. Awwad [4], Nicola Giesen [4,5], Nicola Casiraghi[6,7], Hana Susak [6,7], Katharina Bauer[2,3], Anja Baumann[5], Lukas John [4,5], Anja Seckinger[4,8], Dirk Hose[4,8], Carsten Müller-Tidow [4], Hartmut Goldschmidt [4,9], Oliver Stegle[6,7], Michael Hundemer [4], Niels Weinhold[4,5], Marc S. Raab [4,5,10✉] & Karsten Rippe [1,10✉]

Virtually all patients with multiple myeloma become unresponsive to treatment over time. Relapsed/refractory multiple myeloma (RRMM) is accompanied by the clonal evolution of myeloma cells with heterogeneous genomic aberrations and profound changes of the bone marrow microenvironment (BME). However, the molecular mechanisms that drive drug resistance remain elusive. Here, we analyze the heterogeneous tumor cell population and its complex interaction network with the BME of 20 RRMM patients by single cell RNA-sequencing before/after treatment. Subclones with chromosome 1q-gain express a specific transcriptomic signature and frequently expand during treatment. Furthermore, RRMM cells shape an immune suppressive BME by upregulation of inflammatory cytokines and close interaction with the myeloid compartment. It is characterized by the accumulation of PD1$^+$ γδ T-cells and tumor-associated macrophages as well as the depletion of hematopoietic progenitors. Thus, our study resolves transcriptional features of subclones in RRMM and mechanisms of microenvironmental reprogramming with implications for clinical decision-making.

[1] Division of Chromatin Networks, German Cancer Research Center (DKFZ) and Bioquant, Heidelberg, Germany. [2] Single Cell Open Lab, German Cancer Research Center (DKFZ) and Bioquant, Heidelberg, Germany. [3] Molecular Precision Oncology Program, NCT Heidelberg, Heidelberg, Germany. [4] University Hospital Heidelberg, Internal Medicine V, Heidelberg, Germany. [5] CCU Molecular Hematology/Oncology, German Cancer Research Center (DKFZ), Heidelberg, Germany. [6] Division of Computational Genomics and System Genetics, German Cancer Research Center (DKFZ), Heidelberg, Germany. [7] European Molecular Biology Laboratory (EMBL), Genome Biology Unit, Heidelberg, Germany. [8] Department of Hematology and Immunology, Myeloma Center Brussels, Jette, Belgium. [9] National Center for Tumor Diseases (NCT), Heidelberg, Germany. [10] These authors contributed equally: Marc S. Raab, Karsten Rippe. ✉email: marc.raab@med.uni-heidelberg.de; karsten.rippe@dkfz.de

Multiple myeloma (MM) is a hematological malignancy with clonal expansion of malignant plasma cells in the bone marrow[1]. The current treatment of MM with immunomodulatory drugs (IMiDs), proteasome inhibitors and monoclonal antibodies elicits deep remissions in patients with newly diagnosed MM[2]. However, almost all patients relapse and enter the relapsed/refractory multiple myeloma (RRMM) stage at some point[3]. This disease course is tightly linked to the remarkable genomic complexity of MM that becomes even more pronounced in RRMM and manifests itself as a widespread presence of multiple subclones[4,5]. Furthermore, a number of studies have shown that alterations of non-malignant cells in the bone marrow microenvironment (BME) are critical for the pathogenesis of MM[6–10]. Thus, there is an urgent need to resolve tumor heterogeneity and changes of the BME to reveal molecular patient-specific characteristics that can guide treatment decisions, including emerging T-cell-based immunotherapies. Single-cell RNA sequencing (scRNA-seq) is ideally suited to dissect tumor cell heterogeneity and its microenvironment as shown for MM[11] and cell types in the BME of MM patients[9]. However, an integrated scRNA-seq analysis of both tumor cells and BME from the same patient sample, as well as the application to drug treatment response and RRMM cases are currently lacking.

In this work, we resolve cellular composition, tumor subclone structure and treatment response of RRMM and BME cells from 20 patients. We find that the gain in chromosome 1q (+1q), a high-risk MM aberration that predicts poor prognosis in both newly diagnosed and refractory MM[5,12], emerges from small subclones over treatment lines and shed light on its molecular features. Our study reveals large changes in BME cell-type composition that include (i) an accumulation of exhausted γδ T-cells and myeloid populations with immunosuppressive features, (ii) a depletion of naive T-cells and the B-cell lineage, and (iii) an expansion of reprogrammed plasmacytoid dendritic cells (pDCs) upon treatment with IMiDs. Based on a co-expression analysis of receptor–ligand pairs we provide mechanistic insight on how RRMM cells might reprogram the BME via inflammatory cytokines and ligands of inhibitory receptors and reveal links to +1q.

## Results

We employed single-cell transcriptomics on a droplet-based platform to dissect subclone structure, transcriptional heterogeneity, cellular interactions, and treatment response in RRMM (Fig. 1a–c). The sample set comprised 20 RRMM patients that were refractory to their immediate prior line of treatment with a median of three prior treatments. Of these, 18 patients were refractory to both a proteasome inhibitor and an IMiD, while two patients were primary refractory to initial therapy (Supplementary Tables S1 and S2 and Supplementary Data Set 1). Paired samples before the last treatment and at relapse were analyzed after sorting each sample into a CD138$^+$ plasma/myeloma cell fraction and a CD138$^-$ BME fraction that were processed independently for scRNA-seq analysis (Fig. 1a, b). In total, 212,404 cells were analyzed that passed stringent quality control with a median of 1143 detected genes and 3070 detected UMIs per cell (Supplementary Fig. 1a–f). Two-dimensional embedding using uniform manifold approximation and projection (UMAP) showed that BME immune cells clustered by cell type without the need for further batch-effect correction (Fig. 1c). A clear separation of plasma/myeloma cells from BME immune cells was apparent from the enrichment of CD138$^+$ cells (Supplementary Fig. 1g, h) and the high expression of the plasma cell marker *TNFRSF17* encoding for BCMA. It was further corroborated by automated cell type prediction (Fig. 1c–e), using the Human Cell

Atlas (HCA) bone marrow data set from eight healthy donors as reference. Overall, we profiled 83,201 RRMM plasma cells (PCs) (median = 2189) and 129,203 BME cells (median = 5383) for an integrated analysis of both tumor and immune-cell heterogeneity in RRMM (Supplementary Table 2). Our single-cell transcriptome analysis was combined with clinical data, including type of treatment and depth of response, as well as interphase fluorescence in situ hybridization (iFISH).

**scRNA-seq analysis dissects heterogeneity of RRMM tumor cells.** To assess inter and intra-patient tumor heterogeneity in RRMM we applied clustering and UMAP embedding of RRMM PCs. Strong transcriptional differences between patients were apparent, except for one cluster harboring cells of multiple patients (Fig. 2a and Supplementary Fig. 3a). This cluster could be assigned to non-malignant plasma cells (nPCs) as it had a normal genome based on a copy number alteration (CNA) analysis of the scRNA-seq data, whereas tumor cells of each patient displayed a unique set of numerous chromosomal aberrations (Supplementary Fig. 2a). In addition, nPCs derived from RRMM patients cluster together with nPCs from healthy donors (Supplementary Fig. 2b).

To characterize RRMM subgroups based on their genomic alterations, we first used the iFISH data that include both CNAs and translocations recurrently detected in MM (Fig. 2b and Supplementary Fig. 3b). Pseudo-bulk expression profiles of individual patients followed the iFISH classification and clustered into transcriptional subtypes consistent with their genomic alterations. The marker gene expression profiles included upregulation of *MAFB* and *NSD2* in patients with t(4;14) translocation, increased *CCND1* expression in patients with t(11;14) translocation[13] and increased expression of ribosomal genes in the subset of five hyperdiploid patients[14]. Major CNAs identified from the scRNA-seq data showed a very good agreement (11/12 and 11/13 detected) to those derived from whole-genome sequencing (WGS) data of the same samples (Fig. 2c and Supplementary Fig. 2c). In line with previous studies[5], we identified a high grade of intratumor heterogeneity in RRMM, with a median of three CNA subclones per patient (Supplementary Fig. 2a).

The most frequent CNA in our RRMM cohort was +1q with 17/20 cases. In 10/20 patients +1q was subclonal (Supplementary Fig. 2a). In addition, the fraction of +1q cells identified from scRNA-seq data was highly correlated with the iFISH data (Supplementary Fig. 3c). The only exception was RRMM16 that had a scattered +1q signal, complicating the subclone assignment and indicating a high degree of intratumor +1q heterogeneity (Supplementary Fig. 2a). Thus, the CNA analysis of the scRNA-seq data resolved complex patterns of (rare) subclones and their specific chromosomal aberrations and made it possible to integrate these profiles with the corresponding transcriptomes at single-cell resolution.

The number of clusters per patient increased with the number of cells analyzed. This technical bias made it difficult to assess the correlation between the numbers of clones and clusters (Supplementary Fig. 3d). We generally detected more clusters than clones, which probably reflects non-genetic mechanisms that affect gene expression, including epigenetic alterations and microenvironmental influences. Nevertheless, the gene expression-based clustering frequently showed a high degree of overlap with a given subclone type as illustrated for a patient with five subclones (Fig. 2c–g). In some instances, we also detected rare subclones that did not separate into a distinct transcriptional cluster as depicted for a +1q clone that comprised ~2% of the cells (Fig. 2c, e, f). Differential expression analysis between this

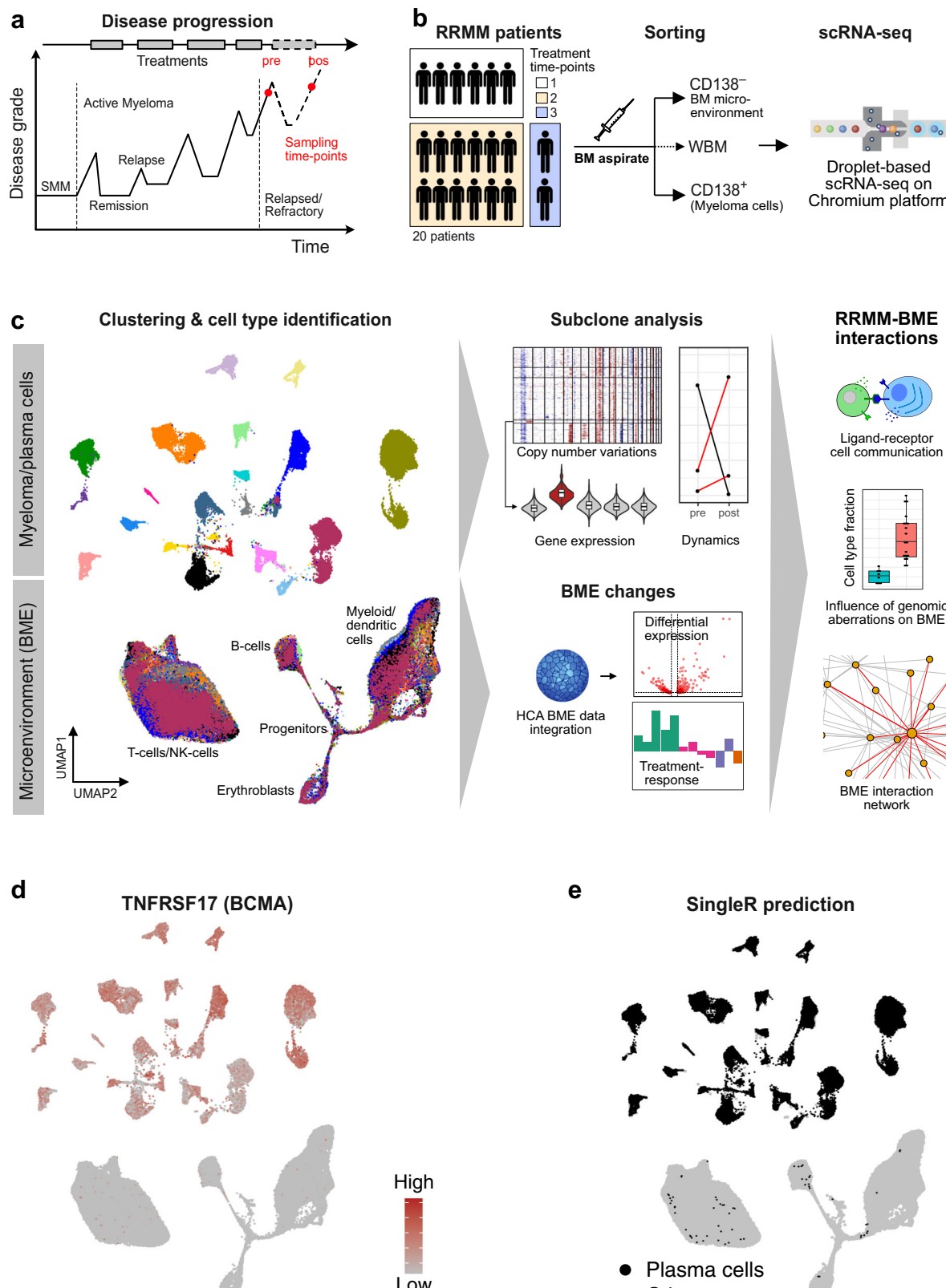

clone and the genetically most similar clone without +1q revealed 328 differentially upregulated genes (Fig. 2h), which were mostly located on 1q (Fig. 2i). These include genes whose increased expression is known to be associated with +1q in MM like

*MCL1*[15], *ATF3*[16], or *PSMD4*[17]. Since +1q is known to be associated with poor prognosis in MM[5,12], we further dissected commonly upregulated genes in +1q subclones to pinpoint the expression of driver genes associated with this genetic aberration.

**Fig. 1 RRMM samples and scRNA-seq data set. a** Typical disease course of RRMM patients and sampling time-points of the study. **b** Workflow for scRNA-seq including CD138 sorting of myeloma cells. **c** Overview of scRNA-seq data analysis. Left: clustering and cell type identification was based on scRNA-seq of 212,404 cells from primary mononuclear bone marrow samples of RRMM patients (n = 20) as shown by an UMAP embedding colored by sample without batch-effect correction. Middle: myeloma cells subclones according to genetic aberrations were identified while changes of BME cells were evaluated against the HCA data set of healthy donors. Right: by integrating these data, subclone-specific interactions of myeloma cells with BME cells were revealed. **d** UMAP embedding as shown in panel (**c**) but colored according to expression of the plasma cell marker *TNFRSF17*. **e** Same as panel (**d**) but colored for plasma cells according to the annotation with SingleR.

**A gene expression signature predicts +1q in single cells.** Based on a differential gene expression analysis across all samples between a +1q clone and the most similar clone without 1q gain we defined a +1q signature based on recurrently upregulated genes (Fig. 2j, k and Supplementary Data Set 2). Furthermore, we classified samples according to their +1q state into "not-detected/rare" (+1q < 10%), "subclonal" (10% > +1q > 80%), and "dominant" (+1q > 80%) (Fig. 2j). The +1q gene expression signature comprised 51 genes and included known drivers of MM pathogenesis that have already been linked to +1q MM such as *ILF2*[16], as well as a number of genes that so far have not been associated with +1q MM like *CTSS* (cathepsin S), a cysteine protease involved in the recruitment of immunosuppressive myeloid cells[18,19] (Fig. 2l and Supplementary Table 4). Next, we validated our +1q signature against a differential expression analysis of a large bulk RNA-seq data set of +1q detected vs. not-detected samples for newly diagnosed MM patients[20]. Two-thirds of +1q signature genes were also detected in the bulk analysis, whereas 18 genes were exclusively detected by scRNA-seq, including *SLAMF7*, *RGS1*, and *CTSS* (Supplementary Fig. 3e, f). In addition, the actin-binding protein *CORO1A* not located on 1q was common in the single-cell and bulk RNA-seq analysis, pointing to trans-regulated downstream effects of +1q on gene expression, as described previously for other genes in MM[11]. Notably, top differentially upregulated genes identified in the bulk RNA-seq approach did not primarily locate to chromosome 1 (Supplementary Fig. 3g). Finally, we validated the +1q signature against an external MM scRNA-seq data set[11] and validated the correlation of +1q signature expression and +1q (sub)clone abundance (Supplementary Fig. 3h, i). Thus, by resolving subclones from CNA analysis and assigning their transcriptional profiles, we derived a +1q gene expression signature. It is based on the differential gene expression analysis of genetically similar subclones with and without +1q in the same sample, which is likely to reduce confounding effects as compared to the bulk RNA-seq analysis.

**Subclones with +1q frequently expand during different treatments.** We next analyzed the behavior of +1q clones during treatment. As an example, an expansion of +1q cells in RRMM13 upon treatment with the second-generation proteasome-inhibitor carfilzomib[21] is shown in Fig. 3a, b. Resistance of myeloma cells to the proteasome-inhibitor bortezomib has been linked to +1q and overexpression of *PSMD4*[22], a recurrently upregulated +1q signature gene. The +1q cells did not separate into a dedicated cluster before treatment based on their complete transcriptome. However, our +1q signature detected a lowly abundant subclone independent of the CNA analysis (Fig. 3c, d). Next, we analyzed the relative abundances of subclones defined by their CNA profile in individual patients over treatment to follow clone dynamics. We detected striking differences between patients, ranging from highly stable compositions to complete rearrangements of clonal distributions (Fig. 3e). Interestingly, we observed a clear association of low clone stability and deeper treatment responses, in line with previous observations in newly diagnosed MM[23]. Importantly, we did not observe a single case where a +1q

subclone became depleted. Rather, +1q clones frequently expanded or remained stable and thus showed a remarkable robustness against different types of treatment in RRMM.

**RRMM is associated with large changes in BME composition.** We integrated the CD138⁻ compartment of RRMM patients with BME cells of healthy donors from HCA into a joint data set of 406,946 cells (Fig. 4a). Fine-grained clustering identified 32 cell types that included all major mononuclear bone marrow cell types and progenitor populations that give rise to myeloid/dendritic, B-cell, and erythroid lineages, as well as disconnected T/NK-cell populations (Fig. 4a, b, Supplementary Fig. 4a, and Supplementary Table 3). Between RRMM patients and healthy individuals, strong differences in cell-type composition were observed, which were mostly in line with previously reported findings (Fig. 4c, d and Supplementary Fig. 4b). These include a depletion of CD4⁺/CD8⁺ naive and CD4⁺ memory T-cells[24] and cells of the B-cell lineage[25]. In addition, CD14⁺ and CD16⁺ monocytes[8,9,26] and effector T-cell populations[25], including CD8⁺ memory and cytotoxic T-cells as well as γδ T-cells were enriched. Interestingly, we found no enrichment of Treg cells and no depletion of GZMK⁺ memory effector T-cells as described previously based on a scRNA-seq analysis for earlier stages of MM[9]. Whereas NK^dim cell frequencies were only slightly higher, immature NK^bright cells increased in abundance. In addition, the hematopoietic progenitor populations were strongly decreased in RRMM (Supplementary Fig. 4b), in line with previous studies[27]. Interestingly, we found this phenomenon to be associated with enhanced immune activation and inflammation-related signaling in the BME of a subset of patients (Supplementary Fig. 4c). Differential expression analysis revealed a strongly enhanced expression of inflammatory cytokines in CD14⁺ monocytes in patients with enhanced inflammation (Supplementary Fig. 4d), indicating that this cell type represents a major driver of inflammation in RRMM. Finally, we observed enhanced expression of the inflammation-induced transcription factor KLF6[28] and its target genes (Supplementary Table 4) across multiple cell types (Supplementary Fig. 4e), indicating that an inflammatory BME induces a common transcriptional program.

**The BME is reprogrammed by RRMM cells via upregulation of inflammatory cytokines.** Reciprocal interactions of tumor and BME cells mediated by cytokines and their corresponding receptors affect several aspects of myeloma pathogenesis including disease progression and treatment resistance[29]. Accordingly, we predicted these cellular interactions based on the expression of ligand–receptor pairs. Across patients, we detected the most pronounced interactions of myeloma cells with the myeloid lineage, in particular with CD14⁺ and CD16⁺ monocytes (Fig. 4e). Monocytes represent an important component of the niche for the homing of nPCs in the bone marrow[30] and myeloma cells to a large extent maintained the same cellular interactions as nPCs in healthy donor samples (Fig. 4f). However, also a diverse set of links was observed (Fig. 4f) that enhanced

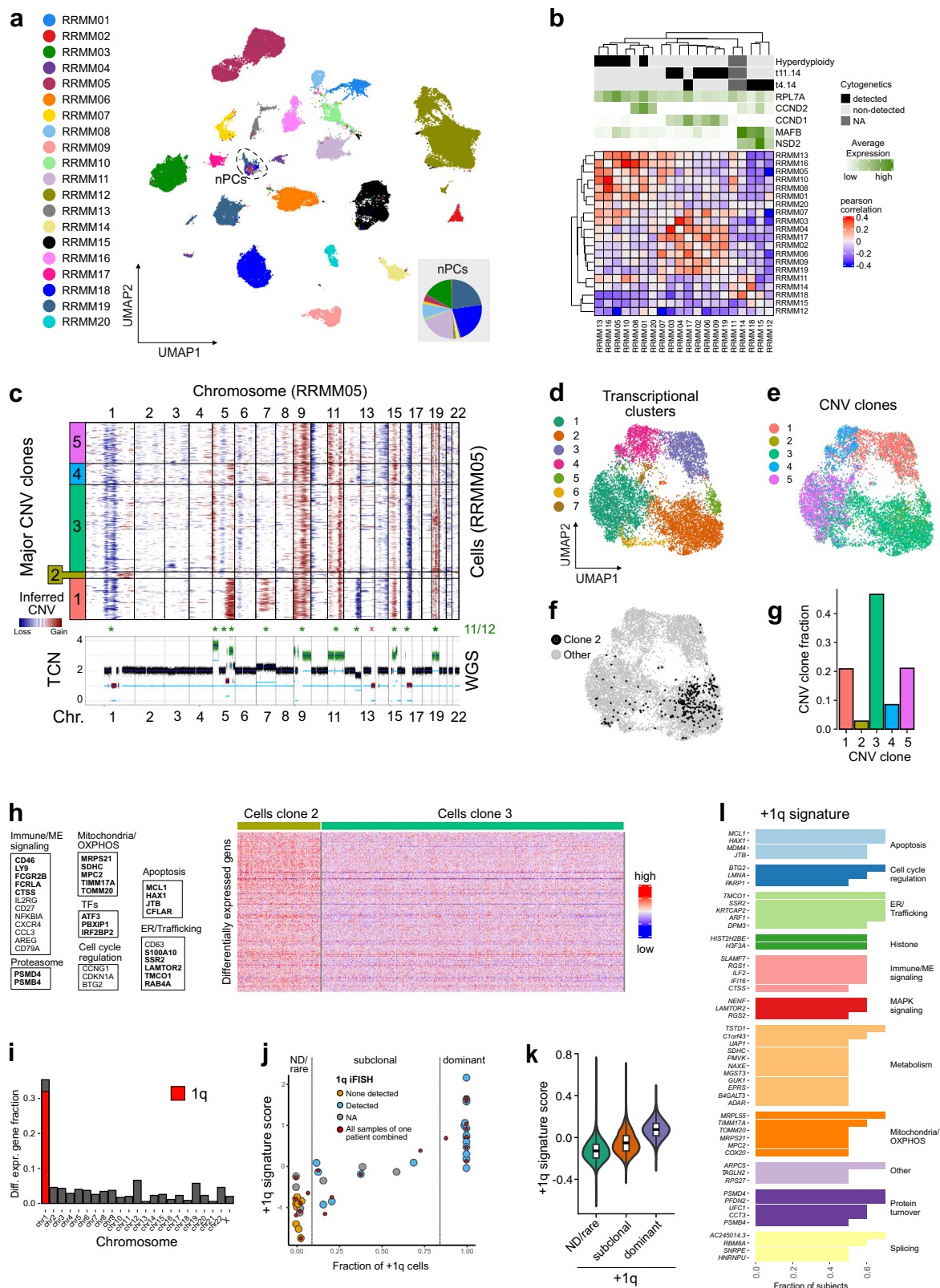

interactions to CD14$^+$ and CD16$^+$ monocytes, cDC2 cells and pDCs (Supplementary Fig. 5a).

We next investigated individual interactions and focused on those that were primarily present in RRMM patients as compared to healthy donors. Several common patterns emerged (Fig. 4g). We observed a frequent upregulation of inflammatory cytokines expressed in myeloma cells. These genes included macrophage migration inhibitory factor (*MIF*), amphiregulin (*AREG*), granulin precursor (*GRN*), and chemokine (C–C motif) ligand 3 (*CCL3*), whose corresponding receptors were primarily expressed in myeloid and dendritic cells. Myeloma cells also frequently upregulated *FAM3C*, a ligand of the inhibitory *KIR2DL3* receptor

**Fig. 2 scRNA-seq analysis of RRMM tumor cells. a** UMAP embedding of RRMM PCs colored by patient. Dashed circle marks non-malignant plasma cells (nPCs). The pie chart inset shows the nPCs fraction colored according to patient. **b** Pearson correlation matrix of averaged gene expression levels per patient. Top, cytogenetic information; bottom, averaged gene expression levels of five MM-subtype-specific genes. **c** CNAs of exemplary patient sample. Top, heatmap of RRMM05 CNA signal normalized against nPCs derived from the HCA bone marrow reference data set. Horizontal lines divide subclones; bottom, coverage plot showing total copy number derived from whole-genome sequencing data of the same sample; * indicates agreement between both modalities in detecting major CNAs. **d** UMAP embedding of RRMM05 tumor cells colored by transcriptional cluster. **e** Same as panel (**d**) but colored according to CNA clone. **f** Same as panel (**d**) but with subclone 2 cells in black. **g** Bar plot of relative subclone abundances in RRMM05. **h** Heatmap of 348 differentially expressed genes between subclone 2 and 3. Example genes are listed and shown in bold letters if located on 1q. Subclone 3 has been downsampled to 1000 cells for visualization. Thresholds for differential expression using two-sided Wilcoxon rank-sum test were $p$-value < 0.05 (Bonferroni-adjusted) and logFC > 0.1. **i** Bar plot showing fraction of differentially upregulated genes in subclone 2 compared to subclone 3 by chromosomal location. **j** Scatter plot of +1q transcriptome signature score against fraction of +1q cells as determined by InferCNV. The Pearson's correlation coefficient was $R = 0.86$ ($p = 2.1 \times 10^{-11}$). Grouping of samples/patients into +1q "not-detected/rare" and "dominant" is indicated by vertical lines. **k** Violin plot of +1q signature scores for cells of the three different +1q groups "ND/rare" ($n = 22,206$ cells), "subclonal" ($n = 13,345$), and "dominant" ($n = 45,946$). The $p$-values from a Kruskal-Vallis test were $< 2 \times 10^{-16}$. Box plot: center line, median; box limits, first and third quartile; whiskers, minimum/maximum values. **l** Genes of the +1q signature. The bar plot shows the fraction of subjects in which individual genes were upregulated in +1q clones when compared to the most similar clone without +1q.

expressed on NK and γδ T-cells[31]. In addition, *CD48* was upregulated in RRMM, which targets the immunomodulatory *CD244* receptor primarily expressed in NK and γδ T-cells[32]. We additionally observed a frequent upregulation of *CD47* in myeloma cells, an integrin-associated receptor protein that inhibits the phagocytosis of target cells[33]. In CD14+ and CD16+ monocytes and in cDC2 cells we observed an upregulation of *CD74* and *CLEC4A* (Supplementary Fig. 5b). *CD74* represents the primary surface receptor for *MIF*, and *CLEC4A* is a regulatory receptor that impairs T-cell immunity[34], underlining the importance of myeloid populations in RRMM pathogenesis. Overall, our scRNA-seq-based interaction analysis revealed a number of transcriptionally overexpressed cytokines and surface markers in myeloma cells that target specific immune cells and thus might contribute to generating an immunosuppressive BME.

**IMiD treatment increases pDCs abundance.** We next systematically analyzed the influence of treatments on BME cell type abundances. B-cells were most severely affected in patients treated with IMiDs (4/5 cases), which likely reflects the lineage relationship of B and plasma cells and associated vulnerabilities to these type of drugs[35] (Supplementary Fig. 6a). In contrast, pDCs expanded upon IMiD-based treatments (5/5 cases), in line with previous studies that have highlighted the importance of pDCs in the pathogenesis of MM, including survival and drug resistance[7]. We therefore investigated transcriptional characteristics of pDCs in RRMM by performing differential expression analysis between pDCs of healthy donors and RRMM patients (Supplementary Fig. 6b, c). The pDCs population in RRMM showed an upregulation of *IRF8*, a transcription factor involved in the differentiation of dendritic cells. Importantly, *IRF8* deletion in pDCs has been shown to increase T-cell stimulatory function[36], indicating that increased *IRF8* expression has the opposite effect. In line with this conclusion, we observed an upregulation of inhibitory receptors *CD300A*[37], *CLEC4A*[34], and *LGALS9*[38] in RRMM pDCs. In addition, pDCs displayed upregulated *TNFSF13B*, encoding for the cytokine BAFF, an important niche factor for plasma cells in the bone marrow[30]. Taken together, our data suggest that a reprogrammed pro-tumorigenic pDC phenotype is important for immunosuppression and IMiD resistance in RRMM.

**γδ T-cells display features of exhaustion in RRMM.** T-cells play a major role in adaptive immunity and are key players in immune surveillance of MM[39]. Therefore, we aimed to identify molecular features of T-cell subsets in RRMM (Fig. 5a–c and Supplementary Fig. 7a) that provide information about functional heterogeneity.

A recent study has provided evidence for the critical role of GZMK+ CD8+ memory effector T-cells and their depletion in earlier stages of MM progression[9]. In contrast, our data set revealed that CD8+ memory effector T-cells became more abundant in RRMM (Supplementary Fig. 4b). Their number further increased upon IMiD-based treatments in 3/7 cases (Supplementary Fig. 6a). Interestingly, we observed a co-expansion of Tregs, indicative of a compensatory mechanism in RRMM (Supplementary Fig. 6a). Furthermore, CD8+ memory effector T-cells exhibited a strong upregulation of activation markers (e.g., *CD69* and *LAT*) and effector molecules (e.g., *GZMB*, *PRF1*, and *GNLY*) in nearly all patients (Supplementary Fig. 7b, c), pointing to a key role of this T-cell type in RRMM immunosurveillance. At the same time, we observed an upregulation of *LAG3*, *KLRG1*, *IFNG*, and *CD47* that have been previously associated with dysfunctional T-cells[40]. This finding might rationalize why activation of GZMK+ CD8+ memory effector T-cells does not result in effective immune responses and clearing of tumor cells in RRMM.

Next, we further examined whether T-cell populations displayed features of exhaustion by upregulation of inhibitory receptors, a phenomenon describing dysfunctional T-cells after chronic antigen stimulation[41]. The highest expression of the exhaustion signature (Supplementary Table 4) was detected in γδ T-cells (Fig. 5d), for which an impaired immune function during MM disease progression has been reported previously[42]. Furthermore, the abundance of this T-cell subset was highly increased in the BME of RRMM patients compared to healthy donors (Supplementary Fig. 4b). When performing a differential expression analysis between γδ T-cells of RRMM patients and healthy donors, we observed a coordinated downregulation of ribosomal genes (Fig. 5e), another characteristic previously linked to T-cell exhaustion[43]. We also detected the significant upregulation of several inhibitory receptors, including *VSIR*, *KLRG1*, *LAG3* and *TIGIT* alongside transcription factors such as *NR4A2* and *ID2* (Fig. 5f) that have been previously linked to T-cell dysfunction[40]. We observed the parallel upregulation of interferon (IFN) response genes (e.g., *IFITM1*, *STAT1*, and *IFI6*) indicating that IFN signaling was associated with the exhausted γδ T-cell phenotype. Expression levels of both exhaustion and IFN-response genes varied across patients (Fig. 5f), suggesting different levels of this γδ T-cell exhaustion phenotype. The grade of exhaustion inversely correlated with the expression of effector genes (Supplementary Table 4), which further supports a dysfunctional phenotype of this T-cell subset in RRMM (Fig. 5g). Interestingly, γδ T-cells in patients with translocation (11;14) were associated with weaker expression of exhaustion genes and

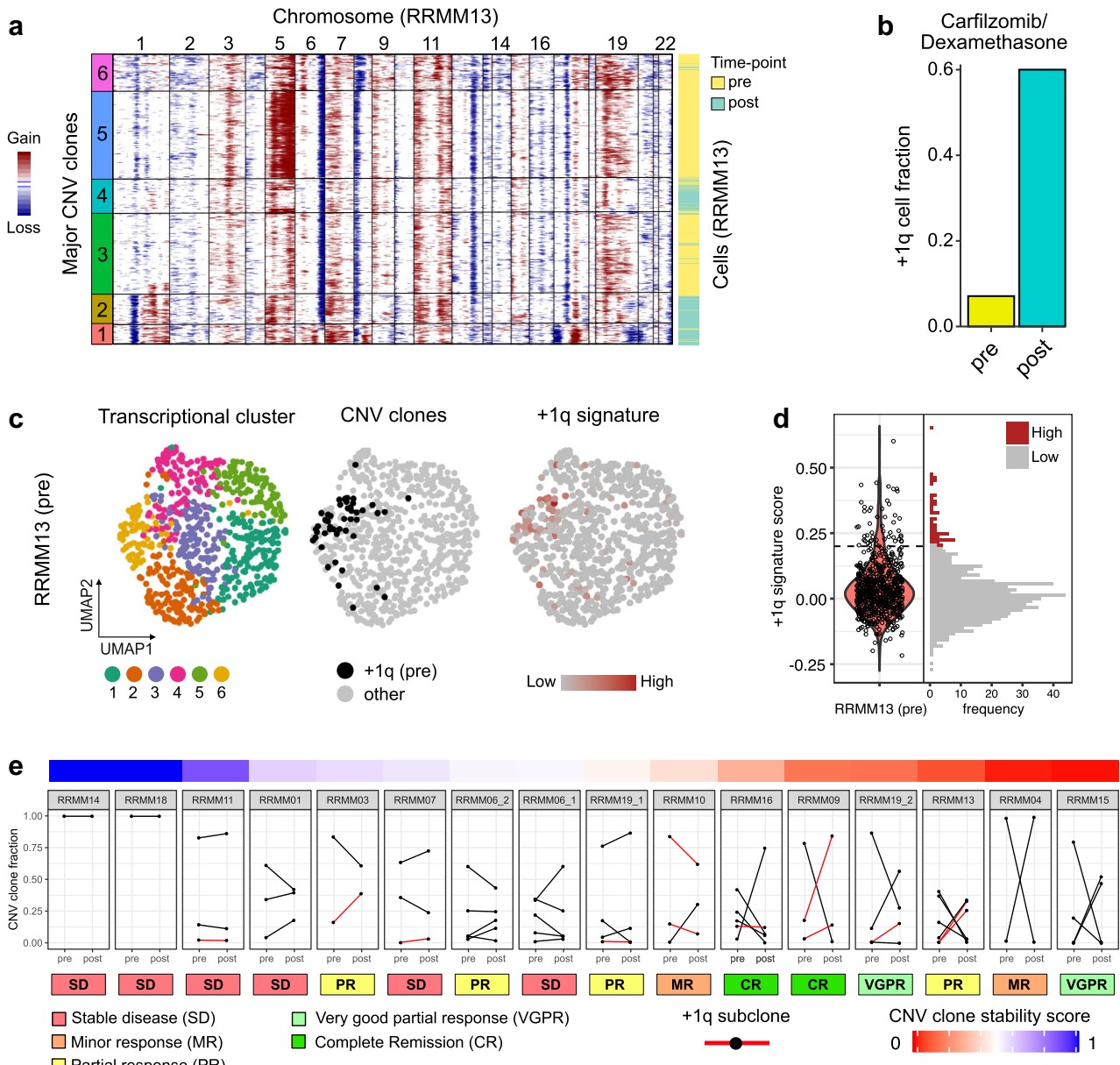

**Fig. 3 Treatment response of +1q subclones. a** Heatmap of RRMM13 CNA signal (pre- and post-treatment data combined). Horizontal lines divide subclones. **b** Bar plot of fraction of +1q cells pre- and post-treatment for RRMM13. **c** UMAP embedding of RRMM13 tumor cells before treatment. Coloring depicts transcriptional cluster (left), +1q cells (middle) and +1q signature score (right). **d** Violin plot and associated histogram of +1q signature score of RRMM13 tumor cells' pretreatment. The positive predictive value (PPV) = 0.076 for the score of 0.2 indicated by the dashed line. **e** Line plot of subclone fraction derived from the CNA analysis per patient pre/post treatment. Red lines mark +1q subclones. Top, CNA stability score per patient; bottom, best treatment response.

higher expression of effector genes. In two patients, we further observed the induction of exhaustion signature genes upon treatment in γδ T-cells (Fig. 5h). As both patients received different types of treatment, this phenomenon might represent a regimen-independent mechanism of immune evasion in RRMM. Using flow cytometry, we validated the increased abundance of γδ T-cells in RRMM and their elevated levels of PD1 (*PDCD1*) compared to CD8+ T-cells in three patients (Supplementary Fig. 7e, f). In contrast, no difference in PD1 expression could be observed between γδ T-cells and CD8+ T-cells in two MGUS (monoclonal gammopathy of undetermined significance) samples, which were used as proxy for healthy individuals. In addition, we confirmed that PD1 expression in γδ T-cells can be increased upon treatment (Supplementary Fig. 7g).

**A macrophage subtype has a myeloma promoting transcription profile**. Next, we characterized tumor-associated macrophages (TAMs) in RRMM as several lines of evidence support their critical role for immunosuppression in MM and other tumor entities[44–46]. The macrophage marker *CD68* was highly expressed in the CD16+ myeloid compartment of RRMM patients (Supplementary Fig. 8a, b), pointing to an infiltration of macrophages into the bone marrow of RRMM patients in line with previous observations[10]. The CD16+ subset displayed a high grade of heterogeneity with 10 subtypes based on different sets of marker genes (Fig. 6a–c). The most abundant subtype with low expression of *CD68* and high expression of *FCGR3A* (encoding for CD16) was assigned to non-classical monocytes (NCM) expressing the glycolytic enzyme *ALDOA*. Five additional lowly abundant NCM types

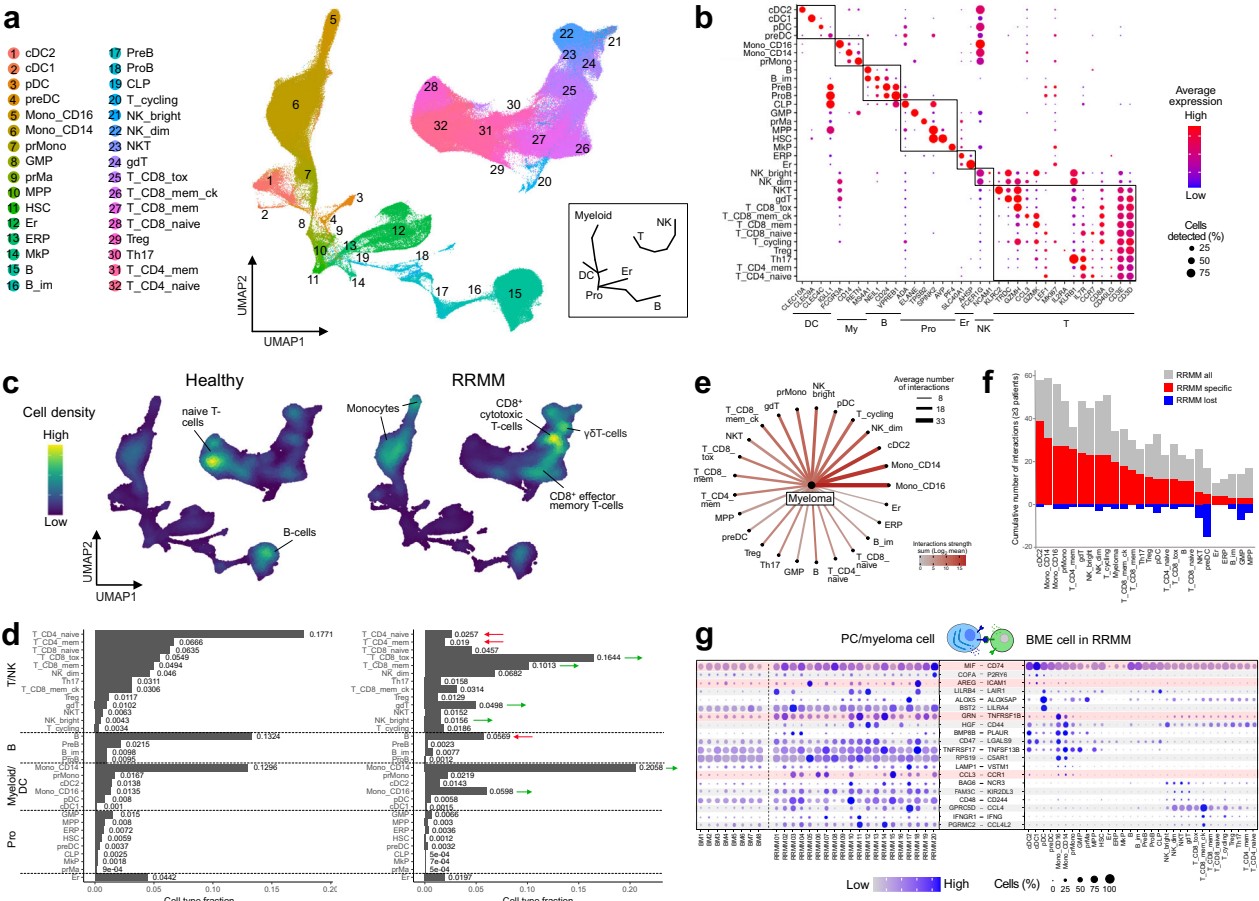

**Fig. 4 Analysis of cellular interaction of myeloma and BME cells. a** UMAP embedding of CD138⁻ BME cells colored by cell type (Supplementary Table 3). The inset provides a schematic overview for the location of major cell types. **b** Gene expression dot plot of major marker genes for individual cell types. **c** UMAP embedding as shown in panel (**a**) as point-density plot split in RRMM vs. healthy donors. **d** Bar plot of cell type fractions for RRMM (right) and healthy (left) donors individually. **e** Cellular interactions of myeloma tumor cells and BME cell types. Ligand–receptor expression was ordered according to the number of detected interactions. **f** Bar plot of the cumulative number of interactions detected in a given BME cell type in RRMM samples in comparison to healthy donors. Gray, all interactions; red, interactions gained in RRMM; blue, interactions lost in RRMM. Only selected interactions detected in ≥3 patients were included. **g** Gene expression dot plot of ligand–receptor expression. Left, expression in nPCs/myeloma tumor cells; right, expression in BME cell types. Only interactions that increased between myeloma tumor cells and immune cell types as compared to nPCs are shown. Interactions involving inflammatory cytokines are highlighted.

were characterized by *CXCL8* (encoding for IL-8), *MEG3, VMO1*, and *LYPD2*[47] as well as *FGD2* expression, a marker of migrating monocytes/macrophages[48]. Most subtypes displayed high expression of *FCGR3A* but two subtypes exhibited a *FCGR3A*^dim phenotype and expressed *CD14*. One of them had lower levels of *CD68* and was therefore annotated as intermediate monocytes (IM). Residual populations with high expression of *CD68* were assigned as TAM1/2/3. A differential expression analysis between NCM and TAM1 subtypes revealed that TAM1 had a high enrichment of IFN-response genes (Supplementary Fig. 8d, e) indicating that these cells represent classically activated M1-like macrophages[49]. Complement genes like *C1QA* were highly expressed in TAM2 and TAM3, whereas TAM3 cells additionally expressed *MRC1* (encoding for CD206), a typical marker for immunosuppressive M2-like TAMs[49]. While NCM were largely depleted, IM and all three TAM populations were strongly enriched in RRMM (Fig. 6d, e). The distinct expression profiles of IM and all three TAM populations suggest different functional roles in the RRMM BME (Fig. 6f). In addition to carrying monocyte markers (e.g., *FCN1*, *VCAN*, and *S100A8*), IM cells expressed genes involved in angiogenesis, including *VEGFA, SELL*, and *HBEGF*. TAM3 cells showed a distinct profile with specific expression of transcription factors

(e.g., *HES1* and *PRDM1*), surface proteins (e.g., *CD163, CLEC10A, FOLR2*, and *ITGAM*) and genes involved in lipid metabolism (e.g., *APOE* and *APOC1*) (Fig. 6f) that have been recently linked to immunosuppressive TAM populations in colorectal and liver cancer[45,46]. In addition, TAM3 cells displayed immunosuppressive features as they preferentially expressed *CD84*, a gene recently linked to myeloid-derived suppressor cells in breast cancer[50], and the negative regulator of T-cell activation *VSIG4*[51]. Interestingly, TAM3 cells showed enhanced expression of *CD38*, a primary drug target in MM[52]. To further characterize the impact of TAMs on the RRMM BME, we constructed an immune cell interaction network that yielded TAMs as nodes of high connectivity (Fig. 6g). They displayed overall higher interaction strengths and connectivities compared to the corresponding populations in healthy individuals (Supplementary Fig. 8g), which is likely to reflect the activated and inflamed BME in RRMM. We next focused on interactions primarily detected in IM/TAM subtypes and identified a specific interaction between TAM3 and NK^bright cells via *IL18* and its receptor (*IL18R1/IL18RAP*) (Fig. 6h). IL18 is a key driver of immunosuppression in MM[53] and can suppress NK-cell activity in cancer[54]. We validated elevated protein expression levels of CD218a (*IL18R1/IL18RAP*) in NK^bright cells, as well as IL-18 and

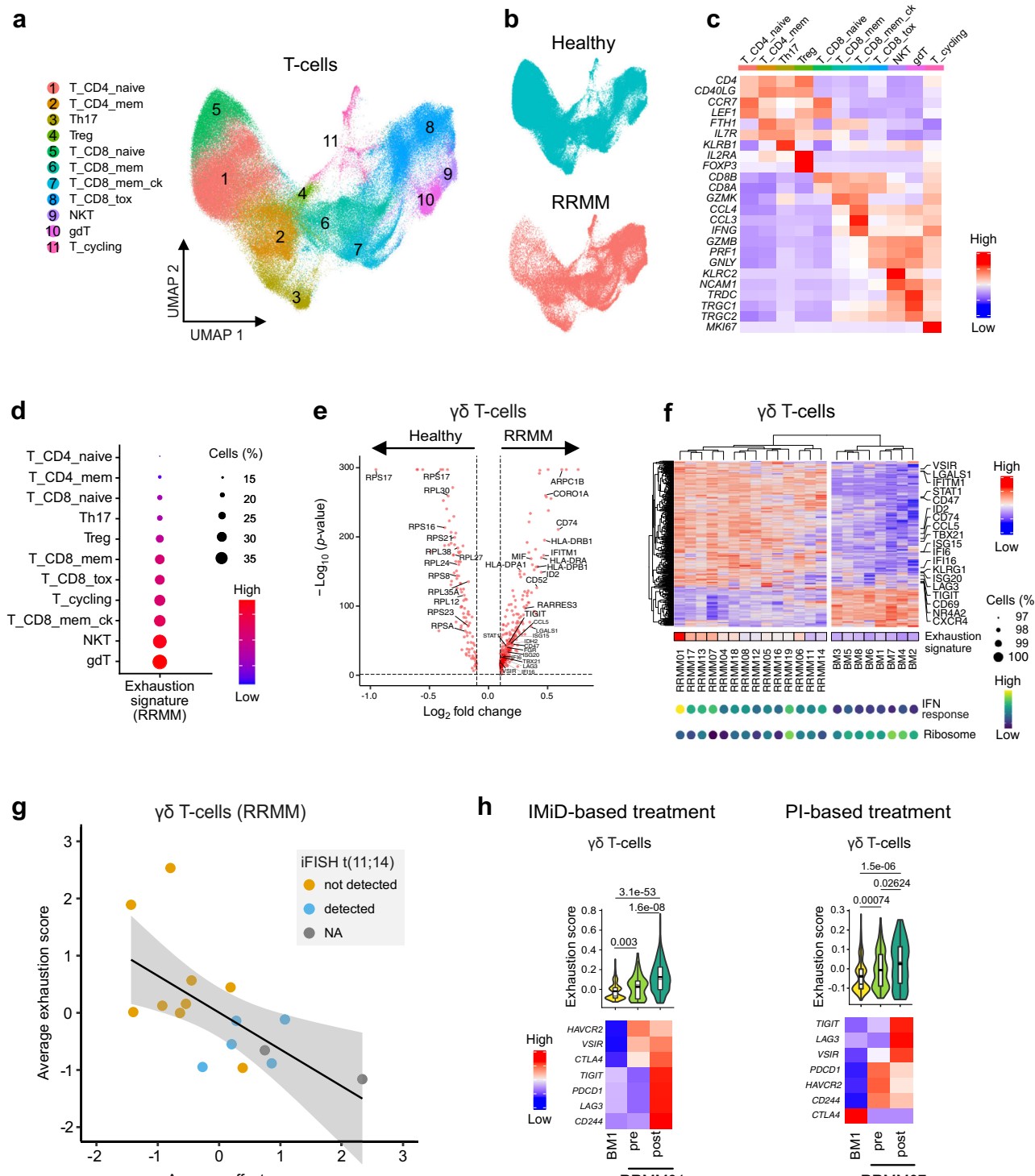

CD38 in TAM3 (CD16+/CD14+/CD11bhi/CD163+) cells by FACS (Supplementary Fig. 9a–c). In contrast, we detected mostly activating interactions of TAM1 and TAM2 with T and NK cells, for example via the expression of *TNFRSF14 – CD160*[55] and *CLEC2B – KLRF1*[56] (Supplementary Fig. 8h). Thus, our results indicate that the TAM1-3 subtypes might exert distinct roles in RRMM.

**TAM3, NK cells, and inflammatory cDC2 abundances are linked to +1q.** Next, we asked whether +1q is linked to specific features of a compromised immune microenvironment.

Interestingly, we found significantly higher numbers of TAM3 and lowered number of NKdim cells in +1q-subclonal/dominant patients when compared to the +1q ND/rare group (Fig. 7a). We next subclustered the NK/NKT populations of RRMM patients (Fig. 7b–d). In addition to CD3+ NKT cells, we identified three NK-cell subtypes: (i) immature NKbright cells characterized by the enhanced expression of *NCAM1*, *KLRC1* (encoding for CD159a/ NKG2A) (Supplementary Fig. 9b), and *SELL* (encoding for CD62L), (ii) activated NKdim cells marked by *CD69*, and (iii) a population that we termed NKdim effector cells, which expressed

**Fig. 5 T-cell heterogeneity in RRMM patient vs. healthy donor samples. a** UMAP embedding of subclustered T-cell populations colored by cell type of the combined RRMM/healthy data set. **b** UMAP embedding split and colored by RRMM/healthy status. **c** Heatmap showing averaged gene expression levels of T-cell population marker genes in the combined data set. **d** Gene expression dot plot showing exhaustion signature score levels in T-cell subpopulations in RRMM. **e** Volcano plot of differentially expressed genes using two-sided Wilcoxon rank-sum test in γδ T-cells (RRMM vs. healthy). Thresholds for differential expression were $p$-value < 0.05 (Bonferroni-adjusted) and logFC > 0.1. **f** Heatmap of clustered average gene expression and exhaustion signature score of γδ T-cells (RRMM vs. healthy). Selected genes are indicated to the right and signature expression levels of IFN response and ribosomal genes are shown at the bottom. Only samples with >60 profiled γδ T-cells were included. **g** Scatterplot of average effector score and average exhaustion score in γδ T-cells across patients. Regression line and 95% confidence interval are shown with data points colored according to iFISH status of t(11;14). The Pearson's correlation coefficient was $R = -0.63$ ($p = 0.0087$). **h** Changes of gene expression in γδ T-cells and CD8[+] cytotoxic T-cells upon treatment with IMiD. Top, violin plot of exhaustion signature score of donor BM1 ($n = 369$ cells) and patients RRMM01 ($n = 65/425$ cells) or RRMM07 ($n = 130/95$ cells) pre- and post-treatment. Pairwise Bonferroni-adjusted $p$-values from a two-sided Wilcoxon rank-sum test are shown. Box plot: center line, median; box limits, first and third quartile; whiskers, minimum/maximum values. Bottom, heatmap of average gene expression levels of genes of the exhaustion signature.

---

high levels of *FCGR3A*, *GZMB*, and *PRF1*. Importantly, NK[dim] effector cells were specifically depleted in +1q patients (Fig. 7d, e). We validated abundance differences of TAM3 and NK[dim] effector subtypes depending on +1q status using flow cytometry analysis in eight samples (Supplementary Fig. 9d).

In addition, changes of a cDC2 subtype were correlated with +1q in myeloma cells after subclustering of cDC2 (CD1c[+]) into three subtypes ("A", "B", and "C") and cycling cDC2 cells (Supplementary Fig. 10a). The cDC2_A-type cells were characterized by enhanced expression of MHC class II genes. In contrast, cDC2_B cells primarily expressed genes involved in the inflammatory response such as S100A8, S100A9, and CD14 (Supplementary Fig. 10b–d) that were also detected in dendritic subtypes in human peripheral blood[57]. Numbers of inflammatory cDC2_B cells were significantly higher in +1q-dominant patients when compared to the ND/rare group (Supplementary Fig. 10e, f). Furthermore, the cDC2_B cells expressed enhanced levels of C-type lectin-like receptors including *CLEC7A*, which encodes for DECTIN1 and is involved in immunosuppression in pancreatic cancer[58], as well as *VSIR* encoding for the inhibitory receptor PD-1H[59] (Supplementary Fig. 10c). Taken together, the analysis revealed a compromised BME that is enriched in pro-tumorigenic and depleted in anti-tumorigenic cell types in +1q RRMM patients.

## Discussion
Our integrative single-cell transcriptome analysis before/after treatment for RRMM dissects intratumor heterogeneity and the interplay between myeloma and BME cells (Fig. 7f). By calling CNAs from the scRNA-seq data, subclones with distinct chromosomal aberrations were identified. This approach allowed us to conduct a differential subclone gene expression analysis within the same sample and to define a +1q signature. It comprised genes involved in a variety of biological processes, including apoptosis, proteasome, and immune/BME signaling, which might explain resistance mechanisms in diverse treatment scenarios. In addition, our signature faithfully detected lowly abundant +1q subclones before treatment, which could be exploited for its application to other MM scRNA-seq data sets. The +1q subclones were remarkably robust against different treatments, in line with previous findings on the association of +1q with inferior outcomes in both newly diagnosed MM and RRMM[5,12].

The RRMM genome and transcriptome profiles obtained in this manner were extended to the analysis of changes of immune cell populations in the BME and the effect of different treatments. The most pronounced BME changes were observed for treatment with IMiDs, which are known to both directly impact myeloma cells and to modulate immune cells towards enhanced anti-MM immunity[60]. In RRMM, we observed an increase of pDCs upon IMiD treatment that displayed a reprogrammed gene expression

profile, suggesting a pro-tumorigenic activity. In line with our observations, previous studies have highlighted the functional role of pDCs in promoting myeloma progression, survival, and drug resistance[7]. Thus, reprogrammed pDCs could represent a key cell type for mediating drug resistance against IMiDs in RRMM.

By dissecting the T-cell population, we found that γδ T-cells were strongly enriched in RRMM compared to healthy individuals and displayed a pronounced exhaustion gene expression signature, which is accompanied by decreased expression of genes involved in effector function. In addition, IFN response genes were upregulated in this cell type. Secretion of type I IFN by myeloma cells has been previously shown to induce Treg expansion and immunosuppression[61], indicating that IFN affects multiple T-cell populations in RRMM. Importantly, the parallel profiling of tumor and immune cells enabled us to predict other direct effects of myeloma cells on γδ T-cells. These include the upregulation of ligands of the inhibitory receptors KIR2DL3 and CD244 by myeloma cells, namely FAM3C[62], which is also a prominent part of the pancreatic cancer secretome[63] and CD48[32]. CD48 has been previously shown to be overexpressed in MM[64] and is involved in mediating inhibitory signaling in the context of T-cell exhaustion[65]. Given the important role of γδ T-cells in recognizing and killing tumor cells in hematopoietic malignancies, we propose to specifically consider γδ T-cells for the generation of cell-based therapies in RRMM[42].

Inflammatory signaling plays a key role in the pathogenesis of MM[66]. It was enhanced in the BME of approximately half of the BME patients studied here. At least partially, it was driven by inflammatory CD14[+] monocytes and induced a common transcriptional program across cell types. This phenomenon was associated with the depletion of hematopoietic progenitor populations, in line with previous observations for IL-1 signaling[67]. In our cellular interaction analysis, myeloma cells displayed an upregulation of inflammatory cytokines (e.g., *CCL3*, *GRN*, *AREG*, and *MIF*) that target receptors primarily expressed in the myeloid and dendritic cell compartment. This phenotype could result from the recently reported extensive upregulation of regulatory elements in NF-κB, NOTCH, and MTOR signaling in myeloma cells compared to normal B-cells[68]. One of the most recurrently upregulated cytokines was MIF whose expression is associated with inferior outcome in MM[69]. Importantly, monocytes are recruited to inflammatory sites where they differentiate into macrophages or dendritic cells[70]. In line with these findings, we report a crucial role of a TAM subtype referred to here as TAM3 in RRMM. The gene expression profile of this M2-like population was strikingly similar to immunosuppressive TAMs described recently for several other tumor entities[45,46,71]. The TAM3 cells expressed inhibitory immune regulators and appeared as a central immune cell interaction hub in our network. In addition, they

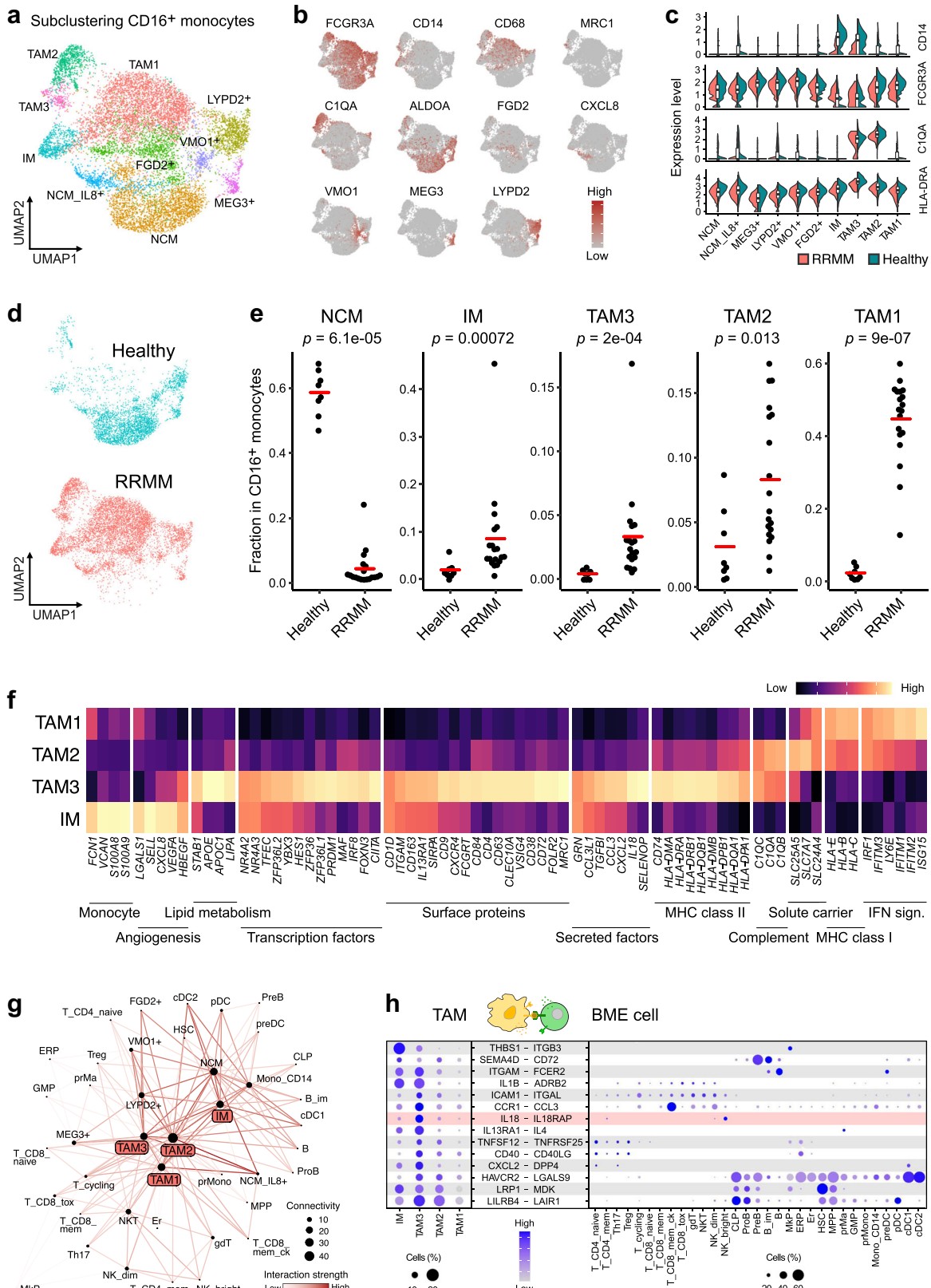

were the primary source of IL18 in RRMM, a key driver of immunosuppression in MM[53]. Our interaction analysis points to a specific IL18-mediated interaction of TAM3 cells with immature NK[bright] cells that exclusively express the IL18 receptor complex. IL18 acts in a context-dependent manner and can mediate both an activation of NK cells[72] as well as a suppression

of NK immunosurveillance[54]. Accordingly, we propose that IL18 secreted by TAM3 cells inhibits the activity of NK[dim] effector cells in RRMM, probably by influencing differentiation of NK[bright] cells towards NK[dim] effector cells. This conclusion is supported by BME characteristics specific for +1q patients who display an increased frequency of TAM3 cells and a depletion of

**Fig. 6 CD16$^+$ monocyte heterogeneity in RRMM. a** UMAP embedding of subclustered CD16$^+$ monocytes colored by subtype. **b** UMAP embedding showing expression levels of major marker genes of subtypes indicated in panel (**a**). **c** Violin plot of selected marker genes of subtypes in panel (**a**) split into RRMM vs. healthy. **d** UMAP embedding as shown in panel (**a**) split and colored by donor status (RRMM/healthy). **e** Beeswarm plot for the comparison of CD16$^+$ monocyte subtype fractions between RRMM ($n = 19$) and healthy ($n = 8$) individuals with Bonferroni-adjusted $p$-values from a two-sided Wilcoxon rank-sum test. Red center line indicates mean. **f** Heatmap showing selected differentially expressed genes between RRMM-enriched CD16$^+$ monocyte subtypes. **g** Network plot of predicted cellular interactions between immune cell subsets in RRMM. Every cell type is connected to its 4 top interacting cell types based on the sum of interaction strengths. The node size corresponds to the number of connected cell types and the coloring corresponds to the interaction strength. **h** Gene expression dot plot of ligand/receptors of RRMM-enriched CD16$^+$ monocyte subtypes (left) and associated interaction partners in the RRMM BME (right). Shown are selected interactions that are stronger in the IM/TAM3 subtypes as compared to TAM2 and TAM3.

NK$^{dim}$ effector cells. The higher TAM3 cell numbers in +1q patients can be linked to our +1q signature, which, for example, shows significantly higher expression levels of cathepsin S (*CTSS*). Cathepsin S is a cysteine protease involved in the recruitment of immunosuppressive myeloid cells in cancer, including M2-like macrophages[18,19]. Thus, it represents a promising candidate gene whose overexpression induces a cascade of BME alterations in RRMM as depicted in the model in Fig. 7f to rationalize how +1q shapes the tumor microenvironment in RRMM.

Our comprehensive data set of both RRMM cells and their microenvironment provides a rich resource that complements previous scRNA-seq studies of patient samples at different stages from smoldering MM to fully developed disease[11] and CD45$^+$/CD138$^-$ cells from the BME[9]. By integrating genetic subclone features and transcriptomes at the single-cell level we introduce an approach to cope with the heterogeneity of RRMM. It dissects different clinically relevant disease subtypes and associated molecular phenotypes and their evolution during treatment in a given patient sample. Thus, the analysis can guide personalized clinical decision making by identifying particularly dangerous subclones and their transcriptome like +1q cells at a very early stage when they comprise only 1–2% of the myeloma cells. Furthermore, the high interpatient tumor heterogeneity, microenvironmental reprogramming and drug response can be evaluated in detail. We anticipate that the insight gained by these approaches introduced in our study for the analysis of RRMM will support the development of novel treatment approaches for the large fraction of MM patients that currently become refractory.

## Methods

**Patient samples**. We studied 20 MM patients that were refractory to their immediate prior line of treatment (Supplementary Table 1). All patients provided written informed consent before participating in the study. Approval was obtained by the ethics committee of the Medical Faculty at the University of Heidelberg. Bone marrow aspirates were 1:1 diluted in preparation buffer (PBS with 0.1% BSA and 2 mM EDTA), and mononuclear cell separation was performed by density centrifugation (Bicoll separating solution, Biochrom) with diluted bone marrow cells (centrifugation 20 min, 1300$g$). Cells were carefully aspirated and washed with preparation buffer (centrifugation 5 min at 470$g$). Red blood cells were lysed using RCL buffer (155 mM NH$_4$Cl, 10 mM KHCO$_3$, 0.1 mM EDTA) for 10 min at room temperature and bone marrow cells were washed (centrifugation 5 min, 470$g$) and resuspended in preparation buffer. After cell counting, $1 \times 10^7$ cells were separated by magnetic activated cell sorting with anti-CD138 microbeads (Miltenyi Biotec) according to the manufacturer's protocol. Relative plasma cell abundance in whole bone marrow (WBM), as well as plasma cell purity of the CD138$^+$ fractions were measured by flow cytometric analysis. Subsequently, up to $2.5 \times 10^6$ cells were frozen in 90% FCS (Sigma-Aldrich) supplemented with 10 % DMSO (Serva Electrophoresis) and stored in liquid nitrogen until further use. For samples with low cell numbers, cells were frozen without sorting for CD138 (WBM). For three patients (RRMM05, RRMM08, and RRMM12) additional samples were also directly processed for scRNA-seq without freezing and/or sorting in order to evaluate the effect of sorting and freezing on data quality and cell-type composition.

**Interphase FISH analyses**. Interphase FISH analysis (iFISH) was performed in a clinical laboratory on 29/36 CD138$^+$ cell fractions as described previously[73] with probes for chromosomal regions 1q21, 5p15, 5q13, 5q35, 6p21, 8p21, 8q24, 9q34, 11q13, 11q22, 11q22.3, 13q14, 14q32, 15q22, 16q23, 17p13, 19q13 and for

translocations t(11;14), t(4;14), t(14;16), or any other IgH- and MYC-rearrangement with unknown region of translocation. Hyperdiploidy was defined as requiring gains of at least two of the three chromosomes 5, 9, and 15. Binary cytogenetic information (detected vs. not-detected) was clustered and visualized with the ComplexHeatmap package using the *dist_letters* function as distance metric (Supplementary Table 5).

**Bulk whole-genome and RNA sequencing**. For bulk whole-genome sequencing (WGS), DNA of CD138-positive plasma cells from two bone marrow samples (RRMM05 and RRMM17) was extracted with the Allprep kit (Qiagen). Saliva was used as germline control and DNA was extracted using the Blackprep Swab DNA kit (Analytik Jena). Libraries were prepared with the Illumina TruSeq Nano DNA kit according to manufacturer's instructions and sequenced on an Illumina HiSeq X system with $2 \times 150$ bp paired-end reads and two lanes per sample at ~80x coverage. Raw sequencing data were processed and aligned using the DKFZ OTP WGS pipeline to human reference genome build 37 version hs37d5. CNAs together with estimation of tumor ploidy and purity were identified using ACEseq. References to the indicated software used for analysis of the WGS data are given in Supplementary Table 5. Bulk RNA-seq data of newly diagnosed MM were from ref. [20] and processed with the nf-core RNA-seq pipeline and using the STAR alignment software. We separated two groups based on 1q21 cytogenetics (1q21-gain detected vs. not-detected) and performed differential expression analysis with DESeq2 using $p_{adj} < 0.05$, log2 fold change (logFC) $> 0.1$, similar to scRNA-seq analysis described below.

**Single-cell RNA sequencing and data preprocessing**. Our protocol used viably frozen cells that were thawed at 37 °C, resuspended in ice-cold PBS and washed twice with cells being collected by centrifugation at 500$g$ for 4 min. The freezing step had little effect on data quality, major cell-type composition and transcriptome as evaluated for three patient samples (Supplementary Fig. 1a–d). Cells were counted with a LUNA automated cell counter (Logos Biosystems). Single-cell capture, reverse transcription, and library preparation were carried out on the Chromium platform (10x Genomics) with the Single Cell 3′ reagent v2 kit (10x Genomics) according to the manufacturer's protocol using 14,000 cells as input per channel. Each of the final libraries were paired-end sequenced (26 and 74 bp) on one Illumina HiSeq 4000 lane. Raw sequencing data were processed and aligned to the human genome (GRCh38) using the CellRanger pipeline (10x Genomics, version 3.01). Bone marrow scRNA-seq raw-count data derived from eight healthy individuals (census of immune cells) were downloaded from the HCA data portal (https://data.humancellatlas.org/explore/projects/cc95ff89-2e68-4a08-a234-480eca21ce79). The latter data set was also generated on the Chromium platform and the Single Cell 3′ reagent v2 protocol as used for our samples. Gene symbols of the HCA data set were converted from GENCODE v27 to v28, which was used throughout our study.

**Quality control of scRNA-seq data**. Cells were excluded from the analysis according to the following criteria: (i) low-quality single-cell libraries with <400 detected genes and >10% mitochondrial counts were removed; (ii) cell doublets were identified with the Scrublet Python package (Supplementary Table 5) using the following parameters: *sim_doublet_ratio* = 2; *n_neighbors* = 30; *expected_doublet_rate* = 0.1. All cells with a doublet score > 0.4 were discarded; (iii) we manually excluded doublet clusters or cells that expressed high levels of marker genes of multiple major bone marrow cell types (e.g., CD3D and HBB) that were not detected by Scrublet; (iv) lowly abundant platelets (PPBP$^+$) and stromal cells (CXCL12$^+$) were excluded; (v) we removed low-quality clusters as defined by high percentages of mitochondrial gene counts, low housekeeping signature scores, and/or low numbers of detected genes, as well as no expression of biologically relevant cell-type- or cell-state-specific genes. In addition, immunoglobulin genes were removed from the analysis due to their extreme abundance in myeloma cells and associated batch effects caused by ambient immunoglobulin mRNAs. Raw counts for immunoglobulin genes and patient/sample-associated information (e.g., sorting fraction, treatment time point, FISH data, etc.) were stored as metadata.

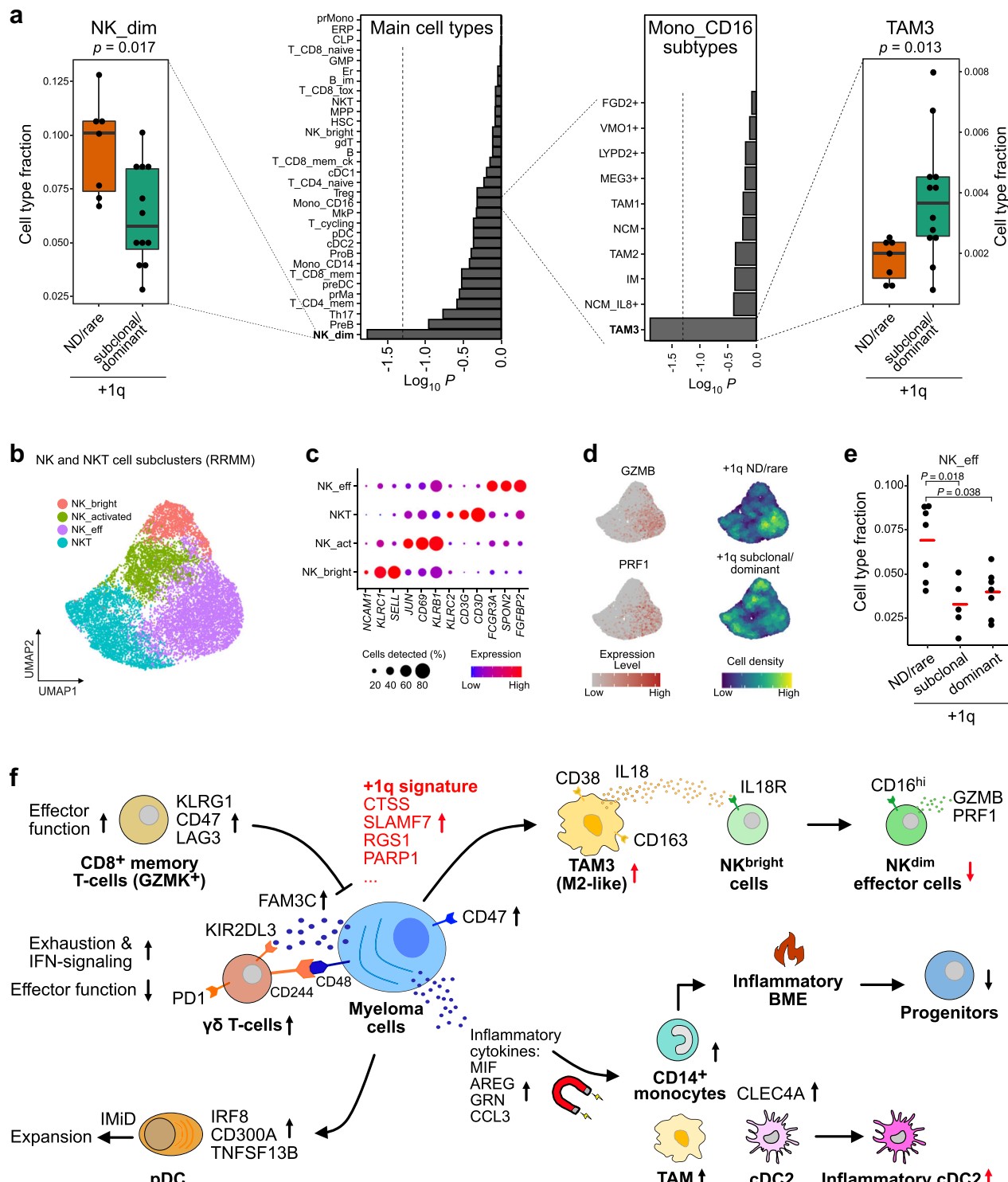

**Fig. 7 BME changes in +1q RRMM. a** Changes in cell-type composition in dependence of +1q. The bar plot in the center shows $\log_{10}$ Bonferroni-adjusted *p*-values from a two-sided Wilcoxon rank-sum test of differences in cell type fractions between +1q ND/rare vs. subclonal/dominant groups. The box plots display the comparison of NK$^{dim}$ cells (left) and TAM3 (right) subtype fractions between patients with +1q ND/rare ($n = 7$) vs. subclonal/dominant ($n = 12$). Box plot: center line, median; box limits, first and third quartile; whiskers, minimum/maximum values. **b** UMAP embedding of subclustered NK cells colored by subtype. **c** Gene expression dot plot for main marker genes of NK subtypes. **d** UMAP embedding showing GZMB and PRF1 expression levels (left) and point-density plot split in +1q ND/rare and +1q-subclonal/dominant groups (right). **e** Beeswarm plot for the comparison of the NK$^{dim}$ effector cells fraction between patients with +1q ND/rare ($n = 7$), subclonal ($n = 5$), and dominant ($n = 7$). Pairwise Bonferroni-adjusted *p*-values from a two-sided Wilcoxon rank-sum test are indicated. Red center line: mean. **f** Scheme of transcriptional changes and altered cellular interactions in RRMM with +1q-specific changes colored in red.

**General scRNA-seq data analysis**. Quality control, normalization, data integration, and unbiased clustering of single-cell transcriptomes was conducted with the Seurat v3 package and the other software listed in Supplementary Table 5. Single-cell RNA-seq data were normalized and highly variable genes were identified using the SCTransform method. Technical or biological confounding effects like mitochondrial counts or cell cycle stages were regressed out using the *vars.to.regress* argument in SCTransform. Principal component analysis (PCA) was applied for linear dimensionality reduction with the top 3000 variable genes. The number of principal components used for downstream clustering was determined with the *ElbowPlot* function in Seurat. For integration of multiple data sets, the reciprocal PCA method implemented in Seurat or Harmony was used with default parameters if not stated otherwise. Reciprocal PCA yielded better results for very large data sets and was used for the HCA BME data set of eight donors with one male and one female donor as reference. Harmony performed better and was more flexible for (multiple) smaller data sets. Shared nearest neighbor graphs were computed using the *FindNeighbors* function in Seurat and cells were clustered using the Louvain algorithm and UMAP embedding in two-dimensional space.

Pseudo-bulk profiles of myeloma scRNA-seq data of individual patients were generated using the *AverageExpression* function in Seurat using scaled normalized counts of the top 3000 variable genes. Pearson correlation coefficients between average expression values were computed with the *cor* function in *R* and clustered with the ComplexHeatmap package. For visualization, diagonal values (=1) were set to 0. Differentially expressed genes between groups of cells were identified by a Wilcoxon rank-sum test ($p_{adj} < 0.05$, logFC > 0.1) using either the *FindMarkers* function in Seurat or the Presto tool. We merged transcriptional clusters if their average gene expression profiles were highly correlated and if they were characterized by similar cell type-specific marker genes. Cell cycle (G1, S, and G2M phase) and signature scores (e.g., housekeeping) were assigned to each cell with the *CellCycleScoring* and *AddModuleScore* function in Seurat. Signature gene lists were derived from the Molecular Signature Database (https://www.gsea-msigdb.org/gsea/index.jsp) or defined manually based on literature (Supplementary Table 4). Gene set enrichment analysis was performed with the hypeR R package. Data were visualized in R using Seurat and other software tools (Supplementary Table 5). For visualization of gene expression differences, we removed a set of 12 non-informative blacklist genes that are highly upregulated across all cell types in MM (*SLC25A6, CD99, IL3RA, CSF2RA, MTRNR2L8, MTRNR2L12, ARL6IP4, POLR2F, GTPBP6, EEF1A1, H3F3A,* and *TMSB4X*). Changes in relative abundances of cell types were calculated using custom code in R. For the analysis of changes in cell-type composition upon treatment, we removed lowly abundant cell types per patient that mainly included progenitor populations and set a threshold of ≥5 cells.

**Cell type annotation and analysis**. Data of all samples were merged and clustered using Seurat. nPCs and myeloma cells were identified based on expressing high levels of *TNFRSF17, SDC1, SLAMF7,* and *CD38*. BME cell types were first annotated individually for the HCA and the RRMM data sets and then combined. The integrated BME data set was subdivided into three major groups, namely T/NK cells, myeloid/DC, and B/erythrocyte lineages plus progenitors. These groups were annotated individually using classical immune cell type marker expression and literature sources according to the description given in Supplementary Table 3. The RRMM BME cell type annotation was corroborated by automated cell type annotation with SingleR (Supplementary Table 5), using the HCA data set and associated cell type labels as reference after downsampling to a maximum of 1000 cells per cell type.

**Analysis of copy number alterations and +1q expression signature**. Copy number changes in individual plasma/myeloma cells were identified using the InferCNV tool (Supplementary Table 5) that averages the expression of adjacent genes over large genomic regions. As reference, we used nPCs derived from the HCA data set and profiled CNAs in myeloma cells of every patient individually as well as all nPCs from the RRMM data set combined. The required gene order file that harbors genomic locations of individual genes was derived from the CellRanger Software (10x Genomics v3.01). For running InferCNV, SCTransform corrected counts were used as input along with the following settings: *cutoff* = 0.1, *cluster_by_groups* = F, *analysis_mode* = "subclusters". The performance of InferCNV to call copy number changes was evaluated by comparing single-cell CNA profiles with bulk WGS and cytogenetic data (iFISH) of the same CD138+ sample. Subclones per patient were identified by cutting the dendrogram of clustered single-cell CNA profiles using the *cutree* function of the dendextend package with a minimum of 40 cells per subclone. Tumor subclone annotations were then mapped to transcriptional profiles using Seurat. In order to identify common genes across patients upregulated in +1q, we performed differential expression analysis (Wilcoxon rank-sum test, *p*-value < 0.05, logFC > 0.1) in individual patients between one +1q clone and its closest relative without 1q gain as determined by InferCNV. Genes (excluding mitochondria encoded genes) that were upregulated in +1q clones in at least 5 out of 10 patients and that were located in the chromosomal 1q region were used to define a +1q signature that comprised 51 genes (Supplementary Table 4). We excluded genes that are not located on 1q (which could indicate for trans-regulation) due to a limited correlation between their

average gene expression and the number of +1q cells across patients. The +1q signature score in single cells was computed with the *AddModuleScore* function in Seurat. Raw-count matrices derived from Ledergor et al.[11] were downloaded from the GEO data base at accession number GSE117156 and patients were grouped into +1q groups according to the CNA analysis in the manuscript. InferCNV analysis of the AL04 donor with subclonal +1q was performed as described above with nPCs derived from the same study. To quantify subclone composition changes upon treatment, we defined a "CNA clone stability score". It uses relative subclone abundances pre- and posttreatment to calculate CNA clone stability scores per patient and treatment according to $\sum(\text{fraction}_{minor}/\text{fraction}_{major})/N$ clones.

**Cellular interaction analysis**. Cellular interactions between cell types were computed based on ligand–receptor co-expression using the CellPhoneDB (v2.0) tool with default settings and log-transformed SCTransform normalized counts as input (Supplementary Table 5). Analysis of interactions between myeloma and immune cells, as well as its comparison with the interactions of nPCs with immune cells in healthy individuals, was conducted for each patient individually. Only cell types with >20 cells per patient were included. To calculate the average number of interactions, we summed all significant (*p*-value < 0.05) interactions between two cell types per patients/donor and averaged this number over patients/donors. To calculate the interactions strength between two cell types across patients, we summed up the mean expression of all ligand and receptor pairs as calculated by CellPhoneDB across patients. For the analysis of interactions of myeloid CD16+ subsets, all cells of a given cell type from all patients/donors were pooled and a network was generated for healthy donors and RRMM patients individually. In this network, cell types are the nodes with their top four interaction partners shown as connecting edges and weighted according to the average strengths and the number of edges.

**Flow cytometry analyses**. For analysis of T-cells, cells were suspended in staining buffer (PBS with 0.5% BSA) and incubated according to the manufacturer's instructions with fluorochrome-labeled antibodies for 30 min at 4 °C, then washed two times. Flow cytometry analyses were performed on a BD FACSLyric flow cytometer; data were analyzed using FlowJo V10.7.1 software (BD). To characterize TAM and NK cells, intracellular staining was performed using a transcription factor buffer set (BD Biosciences) according to the manufacturer's instructions. Briefly, cells were resuspended in PBS and incubated for 15 min at 4 °C with fluorochrome-labeled monoclonal surface antibodies. After washing with 2 ml of PBS, cells were permeabilized with Fix/Perm buffer for 45 min at 4 °C, washed two times with perm/wash buffer, resuspended in an adequate amount of perm/wash buffer, and incubated with intracellular antibodies for 45 min at 4 °C. Subsequently, cells were washed two times with Perm/Wash buffer and resuspended in 0.5% paraformaldehyde in PBS. Flow cytometry analyses were performed on a BD FACSymphony flow cytometer. Data were analyzed using FlowJo V10.7.1 software (BD). All antibodies used in this study are listed in Supplementary Table 6. In addition, the dyes 7-AAD (Becton Dickinson) and Fixable Viability Dye eFluor 506 (ThermoFisher Scientific) were used for cell staining during FACS analysis.

**Reporting summary**. Further information on research design is available in the Nature Research Reporting Summary linked to this article.

## Data availability
The original scRNA-sequencing data sets generated in this study under accession number EGAD00001006903, as well as the four samples additionally analyzed by WGS for validation of the CNA analysis under accession number EGAD00001008150 have been deposited in the European Genome-phenome Archive under the study accession number EGAS00001004805. These data are available under restricted access to comply with German and European data protection regulations, and can be obtained upon application to the linked data access committee. The processed scRNA-seq data used in this study are publicly available at Gene Expression Omnibus under accession number GSE161801. The scRNA-seq data of healthy bone marrow donors were from the Human Cell Atlas database (census of immune cells, [https://data.humancellatlas.org/explore/projects/cc95ff89-2e68-4a08-a234-480eca21ce79]). Bulk RNA-seq raw data from ref. [20] were provided by A.S. and D.H., who can be contacted for access in accordance with German and European data protection regulations. Additional supplementary data sets are listed in Supplementary Table 7. Source data are provided with this paper.

## Code availability
The software used for the data analysis for the different sequencing readouts are listed in Supplementary Table 5. Custom code is provided via a GitHub repository at https://github.com/RippeLab/RRMM[74].

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

## Acknowledgements

We thank the NCT Heidelberg Molecular Precision Oncology Program for technical support and funding through project number HIPO K43R. Further funding was by the Dietmar-Hopp Foundation. We are grateful to Adelheid Cerwenka for discussions, and thank the DKFZ Genomics and Proteomics and Omics IT and Data Management core facilities as well as the SDS@hd scientific data storage of the University of Heidelberg for their help and services.

## Author contributions

Study design and coordination: M.R. and K.R. Acquisition of patient samples: M.R., N.G., A.B. and L.J. Acquisition of data: S.M.T., J.P.M., K.B., M.A., A.S. and D.H. Analysis of data: S.M.T., S.S., A.P., N.C., H.S., O.S. and K.R. Drafting of manuscript: S.M.T. and K.R. Review and editing of manuscript: all authors. Supervision: K.R., M.R., H.G., N.W., C.M.T., M.H. and O.S.

## Funding

## Competing interests

H.G. – Grants/provisions: Amgen, BMS, Celgene, Chugai, Janssen, Johns Hopkins University, Sanofi; Research support: Amgen, BMS, Celgene, Chugai, Janssen, Incyte, Molecular Partners, Merck Sharp and Dohme (MSD), Sanofi, Mundipharma GmbH, Takeda, Novartis; Advisory Boards: Adaptive Biotechnology, Amgen, BMS, Celgene, Janssen, Sanofi, Takeda; Honoraria: Amgen, BMS, Celgene, Chugai, GlaxoSmithKline (GSK), Janssen, Novartis, Sanofi. C.M.T. – Grants/provisions: Pfizer, Daiichi Sankyo, BiolineRx; Research support: Abbvie, Amgen, AstraZeneca, Bayer, Boehringer Ingelheim, Bristol Myers Squibb, Celgene, Eisai, Fresenius, Gilead, Hexal, Janssen, Jazz Pharmaceuticals, MSD, Novartis, Pharmamar, Pfizer, Roche, Shire, Takeda, Affimed; Advisory Boards: Pfizer, Janssen. The other authors declare no competing interests.
