## [Peer review file. · Nature Communications]

REVIEWER COMMENTS

Reviewer #1 (Remarks to the Author): Expert in myeloma genomics

The manuscript submitted by Tirier and coworkers provides a very detailed description of the transcriptome of neoplastic and microenvironmental cells at the single cell level in RRMM. The manuscript confirms previous findings and identifies novel aspects related to this particularly aggressive group of MMs. The data and the findings, although descriptive in nature, represent an important resource for the community and for the design of downstream experiments. The manuscript is well written and documented with informative figures. No further comments from my side.

Reviewer #2 (Remarks to the Author): Expert in multiple myeloma genomics, therapy, and immunogenomics

Comments:

In this study, Tirier et al. for the first time perform a combined analysis of myeloma cells with the immune microenvironment and demonstrate the effect of subclonal/subtype-specific interactions in multiple myeloma with the microenvironment at the level of the single cell transcriptome. The authors generated single-cell RNA-seq data for a total of 212,404 CD138+ myeloma and CD138- BME cells from 20 patients with relapsed/refractory multiple myeloma before and after treatment and resolved cellular composition, subclonal architecture and transcriptional changes in response to treatment.

The authors observe unique copy number profiles for each patient with a high degree of inter- and intra-patient heterogeneity and a median number of 3 subclones per patient and observe good concordance with iFISH/clinical data and lpWGS. The authors identified +1q amplification as the most common genetic alteration and continued to investigate its transcriptional signature by comparing +1q subclones to the most similar genetic subclones without +1q. They identified known genes associated with +1q, as well as unknown genes, including SLAMF7, RGS1 and CTSS. Next, the authors investigated clonal dynamics with treatment and observed variable responses ranging from stable clonal compositions to large differences in clonal structure.

Subsequently, Tirier et al. investigated cell type composition in the BME in RRMM and found compositional differences compared to normal BME, including depletion of CD4+ T cells and enrichment of CD16+ monocytes. The authors predicted cell-cell interactions and identified an increase in inflammatory cytokines produced in MM cells with their corresponding receptors expressed on different immune cell populations. They investigated the effect of treatment on the BME and observe an increase in pDCs upon IMiD treatment which may be linked to resistance. Compared to normal BME, RRMM patients showed increased numbers of gdT cells and increased expression of exhaustion markers. They described further induction of exhaustion signature genes upon IMiD-based treatment in one RRMM patient in gdT and CD8+ cytotoxic T-cells. Furthermore, immunosuppressive macrophage populations are shown to be enriched in RRMM.

Even though this study investigates a heterogeneous cohort of RRMM patients receiving different treatments, this is a valuable dataset of great interest to the community. The authors analyze 212,404 cells from 20 RRMM patients before and after treatment and for the first time perform a combined analysis of myeloma cells with the immune microenvironment. The dataset is of good quality and the analysis is comprehensive. However, there are some outstanding issues that need to be addressed where additional experimental validation and analysis would help. Specifically, the analysis of BME composition is correlative and to establish a causal link with immune activation or response, some functional validation should be performed.

Major:

1. The authors argue that intra-patient heterogeneity on a transcriptional level can largely be explained by CNVs. +1q only leads to upregulation of genes on 1q with little evidence for trans-regulation. As this seems to contradict previous results from Ledergor et al. (2018), the authors should address that. What about transcriptional effects of other frequent subclonal CNVs?

2. Cell-cell interactions are predicted purely on RNA level, missing validation at protein level, which nowadays is feasible. The authors therefore make several assumptions, namely that RNA levels are consistent with protein levels as well as that cells are in physical proximity. Possible ways for validation would include the following:

- o Selected surface marker expression could be validated by flow cytometry of representative primary myeloma samples

- o Similarly, secreted factors like cytokines may be validated by intracellular flow cytometry.

- o Tissue microarrays could be used to show cell interactions in physical space.

3. Interestingly, CD8+ memory effector T-cells are increased and activated in RRMM BME. Can cell-cell interaction analysis give any insights why this does not result in effective immune responses and clearing of MM cells?

4. The authors interestingly observe that the $\gamma\delta$ T-cells and CD8+ cytotoxic T-cells in one RRMM patient (RRMM01) exhibited a strong induction of exhaustion signature genes upon IMiD-based treatment and conclude that IMiD-based treatment may contribute to exhaustion. This is a very interesting finding that should be validated in vitro.

5. The authors detected inhibitory interactions of the tumor-associated macrophages. They report that TAM1/2 interact with T and NK cells via the expression of TNFRSF14 and the CD160 receptor. However, CD160 is an activating receptor in NK cells. KLRF1 is also an activating receptor, meaning that these data are consistent with activation rather than inhibition of NK cells. The authors should therefore provide other evidence for inhibitory interactions if making this claim.

6. The authors show a network with interactions by cell type for RRMM (Fig. 6G). What does this network look like for normal BME?

7. This is the first single cell analysis performed on RRMM samples. The authors should comment on differences in BME composition in RRMM compared to newly diagnosed MM described in Zavidij et al. (2020).

Minor:

8. The authors should provide representative FACS plots for sorted populations.

9. The authors should elaborate on QC filtering. How many cells were sequenced and how many of those passed filtering?

10. The authors should list how many cells from how many normal donors were analyzed even if this data comes from HCA. In plasma cell analysis, normal donors are not included (e.g. in UMAP), do they cluster with nPCs from patients?

11. The presentation of cell-cell interactions (for example Fig. 4G) is hard to follow.

Reviewer #3 (Remarks to the Author): Expert in single-cell RNA-seq and immunogenomics

In the manuscript titled "Subclone specific microenvironmental impact and drug response in refractory multiple myeloma revealed by single cell transcriptomics", Tirier et al present a cohort of 20 bone marrow aspirates from patients with Relapsed/refractory multiple myeloma (RRMM) analysed using single-cell technologies pre- and post- treatment. The authors characterize the malignant cells in terms of CNVs and transcriptional profiles and relate the two. Furthermore, the authors focus on cells with the +1q CNV and examine their expression signature in more detail. They proceed to characterize various aspects of the bone marrow environment.

The authors present the interesting and clinically relevant dataset in great detail and clearly outline many aspects of their analysis, which is presented clearly and compellingly.

There are however some aspects that could benefit from further iteration. Specifically:

- * In Figure 1C the authors show good visual agreement between the CNV profiles detected by single-cells and those detected by WGS. Could the authors quantify this agreement in a more formal way to support the claim that it is "excellent"?

- * I would like to ask the authors to expand on the relationship between CNV and transcriptional differences to more samples. In particular in Figure 2C-2G the authors show very good agreement between the transcriptional and CNV-based detected clusters for a specific sample. It would be of great interest if they could expand this analysis to all samples and specifically address the question of how the CNV similarity of different subclones relates to their transcriptional differences. Furthermore, for each of these clusters it would be interesting to characterize what proportion of transcriptional differences can be directly attributed to loss/gain of gene alleles vs secondary effects.

- * The observation of a +1q transcriptional signature is interesting. Could the authors comment on the PPV of the signature for detecting +1q cells solely on the transcriptomic profile of these cells? The robustness of this transcriptomic signature could benefit by comparing with other external datasets that are likely to harbour +1q cells. I would also suggest that the comparison with bulk that is presented is augmented by the addition of a plot directly comparing fold changes of genes that are included in the signature as the current presentation is unclear.

- * Could the authors outline the existence of other recurrent CNV patterns that are recurrent between different patients and describe any transcriptional signatures they observe common to them?

- * The authors claim that "RRMM cells reprogram the BME by upregulation of inflammatory cytokines". Although the authors provide compelling examples of possible interactions they do not provide experimental evidence of the relevance of these interactions in a biological context. Their claim that these cytokines do contribute to generating an immunosuppressive BME is not fully supported by their data, rather they provide plausible mechanisms by which this may occur. I propose that the text is adjusted so as to reflect these interactions are putative or alternatively that they design further experiments to support their claim.

- * In reference to Figure 6G the authors write that "[...] constructed an immune cell interaction network that yielded TAM3 cells as a central node". The embedding of the TAM3 as a node in the centre of the graph does not on its own constitute evidence of its importance, although it does indeed in this case appear to be a node of high degree. Could the authors comment on the connectivity of this node and the strength of its interactions directly?

- * The analysis of the BME would benefit by scoring individual patient BMEs along specific axis (e.g. cell type composition, inflammatory potential, etc..) and relating this profile to clinical markers at both the pre- post- time point.

* Could the authors comment on the co-variation of the expression of different cell type profiles between patients. Are there any genes that significantly covary between cell types?

Response to reviewer comments for revised manuscript

"Subclone-specific microenvironmental impact and drug response in refractory multiple myeloma revealed by single cell transcriptomics" by Tirier et al.

We are highly grateful to all reviewers for the work that they have put into the critical evaluation of our study and for their specific and constructive suggestions and comments to improve it. We have thoroughly revised our work and have addressed the issues as described in the point-by-point response to the specific comments below (highlighted in blue and renumbered), and feel that these revisions have significantly strengthened our manuscript.

Reviewer #1 (Remarks to the Author): Expert in myeloma genomics

The manuscript submitted by Tirier and coworkers provides a very detailed description of the transcriptome of neoplastic and microenvironmental cells at the single cell level in RRMM. The manuscript confirms previous findings and identifies novel aspects related to this particularly aggressive group of MMs. The data and the findings, although descriptive in nature, represent an important resource for the community and for the design of downstream experiments. The manuscript is well written and documented with informative figures. No further comments from my side.

We thank the reviewer for the very positive evaluation of our work.

Reviewer #2 (Remarks to the Author): Expert in multiple myeloma genomics, therapy, and immunogenomics

In this study, Tirier et al. for the first time perform a combined analysis of myeloma cells with the immune microenvironment and demonstrate the effect of subclonal/subtype-specific interactions in multiple myeloma with the microenvironment at the level of the single cell transcriptome. The authors generated single-cell RNA-seq data for a total of 212,404 CD138+ myeloma and CD138- BME cells from 20 patients with relapsed/refractory multiple myeloma before and after treatment and resolved cellular composition, subclonal architecture and transcriptional changes in response to treatment.

The authors observe unique copy number profiles for each patient with a high degree of inter- and intra-patient heterogeneity and a median number of 3 subclones per patient and observe good concordance with iFISH/clinical data and lpWGS. The authors identified +1q amplification as the most common genetic alteration and continued to investigate its transcriptional signature by comparing +1q subclones to the most similar genetic subclones without +1q. They identified known genes associated with +1q, as well as unknown genes, including SLAMF7, RGS1 and CTSS. Next, the authors investigated clonal dynamics with treatment and observed variable responses ranging from stable clonal compositions to large differences in clonal structure.

Subsequently, Tirier et al. investigated cell type composition in the BME in RRMM and found

compositional differences compared to normal BME, including depletion of CD4+ T cells and enrichment of CD16+ monocytes. The authors predicted cell-cell interactions and identified an increase in inflammatory cytokines produced in MM cells with their corresponding receptors expressed on different immune cell populations. They investigated the effect of treatment on the BME and observe an increase in pDCs upon IMiD treatment which may be linked to resistance. Compared to normal BME, RRMM patients showed increased numbers of gdT cells and increased expression of exhaustion markers. They described further induction of exhaustion signature genes upon IMiD-based treatment in one RRMM patient in gdT and CD8+ cytotoxic T-cells. Furthermore, immunosuppressive macrophage populations are shown to be enriched in RRMM.

Even though this study investigates a heterogeneous cohort of RRMM patients receiving different treatments, this is a valuable dataset of great interest to the community. The authors analyze 212,404 cells from 20 RRMM patients before and after treatment and for the first time perform a combined analysis of myeloma cells with the immune microenvironment. The dataset is of good quality and the analysis is comprehensive.

We thank Reviewer #2 for his/her favorable assessment of our study.

However, there are some outstanding issues that need to be addressed where additional experimental validation and analysis would help. Specifically, the analysis of BME composition is correlative and to establish a causal link with immune activation or response, some functional validation should be performed.

Major:

1. The authors argue that intra-patient heterogeneity on a transcriptional level can largely be explained by CNVs. +1q only leads to upregulation of genes on 1q with little evidence for trans-regulation. As this seems to contradict previous results from Lederger et al. (2018), the authors should address that.

We thank the reviewer for highlighting this important point. In general, we observed some indications for trans-regulation as indicated in Figure S3E and as mentioned in the text for the actin-binding protein CORO1A. However, upon more detailed analysis we observed only a limited correlation between gene expression of genes not located on 1q and the number of cells with 1q gain across patients. Thus, we decided to exclude genes not located on 1q from our signature as including them did not improve its predictive power. We made the following adaptations to the text to clarify this point:

- Results section: *„In addition, the actin-binding protein CORO1A not located on 1q was common in the single cell and bulk RNA-seq analysis, pointing to trans-regulated downstream effects of +1q on gene expression, as described previously for other genes in MM²³.“*
- Methods section: *We excluded genes that are not located on 1q (which could indicate for trans-regulation) due the limited correlation between their average gene expression and the number of cells with 1q gain across patients.*

What about transcriptional effects of other frequent subclonal CNVs?

We agree that transcriptional effects of other subclonal copy number alterations (CNAs) are of high interest (please note that we have replaced the term CNV that refers mostly to endogenously occurring copy number variations with CNA now). However, +1q retains its adverse prognostic impact even in refractory MM¹. Therefore, we have focused our analysis on this genomic alteration. In addition, other recurrent subclonal CNAs were present at a much lower frequency compared to +1q in our dataset, which precludes computing a robust signature for them. Nevertheless, we believe that our approach is generally applicable to all types of CNAs that can be called from the scRNA-seq data and anticipate that it will be of great interest for other future studies, e.g., in newly diagnosed MM.

2. Cell-cell interactions are predicted purely on RNA level, missing validation at protein level, which nowadays is feasible. The authors therefore make several assumptions, namely that RNA levels are consistent with protein levels as well as that cells are in physical proximity. Possible ways for validation would include the following:

- Selected surface marker expression could be validated by flow cytometry of representative primary myeloma samples
- Similarly, secreted factors like cytokines may be validated by intracellular flow cytometry.
- Tissue microarrays could be used to show cell interactions in physical space.

We fully agree with the reviewer that it is important to validate our RNA based analysis on the protein level. We have addressed this issue with two sets of flow cytometry validation experiments and found a very good agreement between the scRNA-seq and FACS data. Thus, we conclude that the scRNA-seq centric approach used in our study provides reliable information on the changes in cell type abundance and functional features inferred from gene expression in RRMM. Specifically, we conducted the following FACS analysis that is now briefly described in the main text and included in the new supplementary **Figures S7 and S9**.

- We designed a FACS panel to identify NK^{dim} effector (CD56^{dim}/CD16^{Hi}/GZMB⁺) and TAM3 (CD16⁺/CD14⁺/CD11b^{Hi}/CD163⁺) cells and validated their abundance in dependence of +1q status. In addition, we validated enhanced expression of IL18, CD38 (TAM3) as well as IL18R and CD159a (NK^{dim} effector cells) using the same panel as validation for protein expression and cellular interactions. The associated results are presented in **Figure S9**.
- As we observed the highest transcript levels of T-cell exhaustion-related inhibitory receptors in $\gamma\delta$ T-cells, we decided to focus on validating this finding also on the protein level by FACS. Importantly, we could validate elevated levels of PD1 in $\gamma\delta$ T-cells compared to CD8⁺ T-cells in three patients, whereas no difference in PD1 expression could be observed between $\gamma\delta$ T-cells and CD8⁺ T-cells in two MGUS samples. We added the associated results to **Figure S7D-F**.

We also rephrased the paragraphs and paragraph titles that relate to the analysis to predict cellular interactions to point out that these results represent predictions and are based on transcriptional measurements.

With respect to tissue microarrays, we agree that these, as well as spatial transcriptomics approaches, would also be informative to validate our conclusions on cellular interactions.

However, the bone marrow represents one of the most difficult tissues in terms of handling and histological processing, and established protocols for the applications of the above methods to human bone marrow at single cell resolution are currently lacking. Thus, we have refrained from taking this route as we were successful to validate important conclusions from our scRNA-seq analysis by the FACS experiments.

3. Interestingly, CD8⁺ memory effector T-cells are increased and activated in RRMM BME. Can cell-cell interaction analysis give any insights why this does not result in effective immune responses and clearing of MM cells?

In order to address this point, we re-evaluated differentially expressed genes between GZMK⁺ CD8⁺ memory effector T-cells derived from healthy donors and RRMM patients. This analysis yielded a number of promising candidate genes including LAG3, KLRG1, IFNG and CD47 that have been previously associated with dysfunctional T-cells and immune-suppression. We also conducted an analysis of cellular interactions, but this did not yield further insight. We adapted Figure S7B and S7C accordingly and added the following paragraph to the manuscript:

“At the same time, we observed an upregulation of LAG3, KLRG1, IFNG and CD47 that have been previously associated with dysfunctional T-cells² and immunosuppression³. This finding might rationalize why activation of GZMK⁺ CD8⁺ memory effector T-cells does not result in effective immune responses and clearing of tumor cells in RRMM.”

4. The authors interestingly observe that the $\gamma\delta$ T-cells and CD8⁺ cytotoxic T-cells in one RRMM patient (RRMM01) exhibited a strong induction of exhaustion signature genes upon IMiD-based treatment and conclude that IMiD-based treatment may contribute to exhaustion. This is a very interesting finding that should be validated in vitro.

We analyzed the induction of exhaustion in $\gamma\delta$ T-cells upon treatment in further detail. Interestingly, we found another case where expression levels of exhaustion genes are significantly elevated post-treatment, albeit less pronounced. However, this patient underwent a PI-based therapy, thereby questioning our initial hypothesis. We added this 2nd case to Figure 5H and rephrased the associated paragraph as following: *“As both patients received different types of treatment, this phenomenon might represent a regimen-independent mechanism of immune evasion in RRMM. Using fluorescence activated cell sorting (FACS), we validated the increased abundance of $\gamma\delta$ T-cells in RRMM and their elevated levels of PD1 (PDCD1) compared to CD8⁺ T-cells in three patients (Figure S7E and S7F).”*

We also removed associated results for CD8⁺ effector T-cells from **Figure 5** in order to elaborate in further detail on the results obtained for $\gamma\delta$ T-cells. In addition, as mentioned above, we validated the elevated PD1 expression in $\gamma\delta$ T-cells upon treatment in patient RRMM01 by FACS. We added the associated results to **Figure S7F**.

5. The authors detected inhibitory interactions of the tumor-associated macrophages. They report that TAM1/2 interact with T and NK cells via the expression of TNFRSF14 and the CD160 receptor. However, CD160 is an activating receptor in NK cells. KLRF1 is also an activating receptor, meaning that these data are consistent with activation rather than inhibition

of NK cells. The authors should therefore provide other evidence for inhibitory interactions if making this claim.

We thank the reviewer for pointing out this issue. Our initial hypothesis was based on the study by Cai et al. ⁴, which showed that CD160 can inhibit the activation of human CD4+ T cells through its interaction with TNFRSF14. However, CD160 is rather expressed in CD8+ T-cells in our RRMM dataset. After a thorough literature search, we thus agree that these interactions are more likely to have an activating role. We rephrased the associated paragraph accordingly: *“In contrast, we detected mostly activating interactions of TAM1 and TAM2 with T and NK cells, for example via the expression of TNFRSF14 – CD160 ⁵ and CLEC2B – KLRF1 ⁶ (Figure S8H). Thus, our results indicate that the TAM1-3 subtypes might exert distinct roles in RRMM.”*

6. The authors show a network with interactions by cell type for RRMM (Fig. 6G). What does this network look like for normal BME?

We also generated a network for healthy individuals which is now included as **Figure S8G**. We also adapted the network visualization so that the node size reflects the number of connections and the edge width corresponds to the interaction strength between two cell types (= sum of interaction strength).

7. This is the first single cell analysis performed on RRMM samples. The authors should comment on differences in BME composition in RRMM compared to newly diagnosed MM described in Zavidij et al. (2020).

We report two key differences we find in RRMM in comparison to the Zavidij et al. study ⁷. One aspect was already included in the initial manuscript, which states that “A recent study has provided evidence for the critical role of GZMK⁺ CD8⁺ memory effector T-cells and their depletion in earlier stages of MM progression ⁷. In contrast, our dataset revealed that CD8⁺ memory effector T-cells became more abundant in RRMM (**Figure S4B**)”. Furthermore, in the paragraph “RRMM is associated with large changes in cell type composition”, we have slightly rephrased the previous statement to ref. ⁷ to clarify that is also based on scRNA-seq: *„Interestingly, we found no enrichment of Treg cells and no depletion of GZMK⁺ memory effector T-cells as described previously based on a scRNA-seq analysis of earlier stages of MM ⁷“*. Thus, we feel that we have covered the main differences we find in our analysis of RRMM as compared to other MM stages analyzed in the previous study.

Minor:

8. The authors should provide representative FACS plots for sorted populations.

Sorting has been done with magnetic beads, thus no FACS plots are available.

9. The authors should elaborate on QC filtering. How many cells were sequenced and how many of those passed filtering?

QC parameters for scRNA-seq data are described in the methods section "Quality control of scRNA-seq data". Additionally, we added cell numbers before filtering (CellRanger output) and after QC per sample to Sup_Data_01_samples.xlsx.

10. The authors should list how many cells from how many normal donors were analyzed even if this data comes from HCA.

We added this information to **Table S2**.

In plasma cell analysis, normal donors are not included (e.g. in UMAP), do they cluster with nPCs from patients?

Yes, they do. We added two UMAP plots to **Figure S2** where nPCs from healthy donors are clustered together with tumor cells and nPCs from RRMM patients.

11. The presentation of cell-cell interactions (for example Fig. 4G) is hard to follow.

Similar to **Figure 6H**, we have added colored horizontal lines to highlight interactions involving inflammatory cytokines and to emphasize how these plots should be read.

Reviewer #3 (Remarks to the Author): Expert in single-cell RNA-seq and immunogenomics

In the manuscript titled "Subclone specific microenvironmental impact and drug response in refractory multiple myeloma revealed by single cell transcriptomics", Tirier et al present a cohort of 20 bone marrow aspirates from patients with Relapsed/refractory multiple myeloma (RRMM) analysed using single-cell technologies pre- and post- treatment. The authors characterize the malignant cells in terms of CNVs and transcriptional profiles and relate the two. Furthermore, the authors focus on cells with the +1q CNV and examine their expression signature in more detail. They proceed to characterize various aspects of the bone marrow environment.

The authors present the interesting and clinically relevant dataset in great detail and clearly outline many aspects of their analysis, which is presented clearly and compellingly.

We thank Reviewer #3 for the very positive assessment of our work.

There are however some aspects that could benefit from further iteration. Specifically:

* In Figure 1C the authors show good visual agreement between the CNV profiles detected by single-cells and those detected by WGS. Could the authors quantify this agreement in a more formal way to support the claim that it is "excellent"?

We thank the reviewer for raising this point. We quantified the agreement in CNV calling between WGS and scRNA-seq (InferCNV) by counting the major CNAs detected by scRNA-seq that have been detected by WGS. We performed this analysis for one additional sample

(**Figure S2C**). In both cases the vast majority of major CNVs detected by WGS were also detected by InferCNV (11/12 and 11/13). We added the agreement information to both **Figure 2C and S2C**. We also partially rephrased the paragraph that describes this finding: „Major copy number alterations derived from the scRNA-seq data show a very good agreement (11/12 and 11/13 detected) to those identified from whole genome sequencing (WGS) data of the same samples (**Figure 2C and Figure S2C**).“

* I would like to ask the authors to expand on the relationship between CNV and transcriptional differences to more samples. In particular in Figure 2C-2G the authors show very good agreement between the transcriptional and CNV-based detected clusters for a specific sample. It would be of great interest if they could expand this analysis to all samples and specifically address the question of how the CNV similarity of different subclones relates to their transcriptional differences.

In order to address this point, we analyzed the relationship between the number of profiled cells, transcriptional clusters and CNV clones. While the number of cells and clusters per patient was highly correlated, we observed an overall weak correlation between the numbers of clones and clusters (**Figure S3D**). However, numbers of cells and clones deviate only minimally in most patients (**Figure S3D**). We generally detected more clusters than clones which probably reflects non-genetic mechanisms that affect gene expression, including epigenetic alterations and microenvironmental influences.

Furthermore, for each of these clusters it would be interesting to characterize what proportion of transcriptional differences can be directly attributed to loss/gain of gene alleles vs secondary effects.

We agree with the reviewer that a detailed analysis of how transcriptional differences can be directly attributed to loss/gain of gene alleles vs. secondary effects would be highly interesting and very valuable for the field. However, we think that this kind of analysis is not a simple extension of the work we already did, but would require a different approach and a significant amount of additional analysis work with unclear outcome. One bottleneck here is the resolution of detecting genomic aberrations from the scRNA-seq data that requires sufficiently large genomic regions to be binned infer CNAs from expression differences. Furthermore, while we have a sufficient number of samples with/without +1q, this is not the case for loss/gain of genes in general. Therefore, we have refrained from conducting this analysis.

* The observation of a +1q transcriptional signature is interesting. Could the authors comment on the PPV of the signature for detecting +1q cells solely on the transcriptomic profile of these cells?

With InferCNV results as ground truth, we calculated PPV = 0.76 for +1q “high” cells as detected by the +1q signature. We added this information to Figure 3D.

The robustness of this transcriptomic signature could benefit by comparing with other external datasets that are likely to harbour +1q cells.

We followed up on this suggestion and used the scRNA-seq dataset of Ledergor et al. ⁸ to replicate **Figure 2K** with an external data. We grouped patients in the three +1q categories (ND/rare, subclonal, dominant) according to Figure 2D in Ledergor et al. ⁸. We added the associated plot as Figure S3H. We also analyzed one individual patient (AL04) to validate subclonal expression of the +1q signature. We added the associated plot as Figure S3I-J. The results show that our +1q signature works well even with a data set that has a much lower number of cells and less genes per cell.

I would also suggest that the comparison with bulk that is presented is augmented by the addition of a plot directly comparing fold changes of genes that are included in the signature as the current presentation is unclear.

We changed the Volcano-plot describing differential expression of bulk RNA-Seq data by highlighting genes of the +1q signature identified by scRNA-seq (**Figure S3F**).

* Could the authors outline the existence of other recurrent CNV patterns that are recurrent between different patients and describe any transcriptional signatures they observe common to them?

See above comment to major point #1 raised by Reviewer #2, which is repeated here again: We agree that transcriptional effects of other subclonal copy number alterations (CNAs) are of high interest (please note that we have replaced the term CNV that refers mostly to endogenously occurring copy number variations with CNA now). However, +1q retains its adverse prognostic impact even in refractory MM ¹. Therefore, we have focused our analysis on this genomic alteration. In addition, other recurrent subclonal CNAs were present at a much lower frequency compared to +1q in our dataset, which precludes computing a robust signature for them. Nevertheless, we believe that our approach is generally applicable to all types of CNAs that can be called from the scRNA-seq data and anticipate that it will be of great interest for other future studies, e.g., in newly diagnosed MM.

* The authors claim that "RRMM cells reprogram the BME by upregulation of inflammatory cytokines". Although the authors provide compelling examples of possible interactions they do not provide experimental evidence of the relevance of these interactions in a biological context. Their claim that these cytokines do contribute to generating an immunosuppressive BME is not fully supported by their data, rather they provide plausible mechanisms by which this may occur. I propose that the text is adjusted so as to reflect these interactions are putative or alternatively that they design further experiments to support their claim.

We agree with the reviewer that we need to tone down some of our statements and have done so in the revised manuscript. We have rephrased the abstract as well as the associated paragraph in the Results section to point out that we present predictions that are based on transcriptional measurements. We also addressed the reviewer's comment with the newly added flow cytometry validation experiment for TAM3-NK cell interactions. We designed a FACS panel to identify NK^{dim} effector (CD56^{dim}/CD16^{Hi}/GZMB⁺) and TAM3 (CD16⁺/CD14⁺/CD11b^{Hi}/CD163⁺) cells and validated their abundance in dependence of +1q status. In addition, we validated enhanced expression of IL18, CD38 (TAM3) as well as IL18R

and CD159a (NK^{dim} effector cells) using the same panel as validation for protein expression and cellular interactions. The associated results are presented in **Figure S9**.

* In reference to Figure 6G the authors write that "[...] constructed an immune cell interaction network that yielded TAM3 cells as a central node". The embedding of the TAM3 as a node in the centre of the graph does not on its own constitute evidence of its importance, although it does indeed in this case appear to be a node of high degree. Could the authors comment on the connectivity of this node and the strength of its interactions directly?

We adapted the network visualization so that the node size reflects the number of connections and the edge width corresponds to the interaction strength between two cell types (= sum of interaction strength) (**Figure S8G**). We also rephrased the associated paragraph: "*To further characterize the impact of TAMs on the RRMM BME, we constructed an immune cell interaction network that yielded TAMs as nodes of high connectivity (Figure 6G) with overall higher interaction strengths and connectivities compared to healthy individuals (Figure S8G), which likely reflects the activated and inflamed BME in the diseased state.*"

* The analysis of the BME would benefit by scoring individual patient BMEs along specific axis (e.g., cell type composition, inflammatory potential, etc..) and relating this profile to clinical markers at both the pre- post- time point.

To address this question, we scored BMEs of individual patients for pathways associated with immune activation/inflammation. Interestingly, we found a clear association between the relative abundance of hematopoietic progenitor populations and immune activation/inflammation-related signaling pathways. We grouped patients according to the grade of immune cell activation/inflammation ("high", "intermediate", "low"). We could not find any evidence that treatments influence this observation. We added the associated figure panel to **Figure S4**.

We could also show an inverse correlation between effectoriness and exhaustion in gdT cells in which patients with t(11.14) appear to be rather associated with high effector function and lower levels of exhaustion. We added the associated figure panel to **Figure 5G**.

* Could the authors comment on the co-variation of the expression of different cell type profiles between patients. Are there any genes that significantly covary between cell types?

We have conducted an analysis that is similar to the one shown in **Figure 4** of Zavidij et al. ⁷. In the latter study, the authors show that genes involved in interferon-related signaling co-vary in different cell types, especially monocytes and T-cells, and that this co-variation can be attributed to different patients/donors. Accordingly, we further examined genes expression differences according to the grade of immune cell activation/inflammation ("high", "intermediate", "low") in individual cell types (**Figure S4C**). First, we identified CD14⁺ monocytes as a potential driver of inflammation as they expressed high levels of inflammatory cytokines specifically in patients with "high" immune cell activation/inflammation (**Figure S4D**). In addition, we observed an upregulation of the inflammation-induced transcription factor KLF6 as well as its target genes across cell types in patients with "high" immune cell

activation/inflammation (**Figure S4E**), indicating that an inflammatory BME induces a common transcriptional program. These findings are now included in the revised manuscript in **Figure S4C-E** and the associated text.

Additional changes made in the revised manuscript

- We optimized the UMAP of the bone marrow microenvironment in **Figure 4A**. In the previous version cycling T-cells primarily clustered with B-cell progenitors due to cell cycle influence.
- We optimized the coloring of the interaction plot **Figure 4E**.
- We updated the scheme in **Figure 7F** to account for the additional findings that were made during the revision of our study.
- We have replaced the term “copy number variation” (CNV), which refers mostly to endogenously occurring copy number differences with “copy number alteration” (CNA) that refers to cancer-related genomic changes in copy number.
- Mohamed Awwad, Michael Hundemer, Anja Seckinger and Dirk Hose have been added as authors due to their contributions to the revisions of the manuscript.

References

- 1 Ziccheddu, B. *et al.* Integrative analysis of the genomic and transcriptomic landscape of double-refractory multiple myeloma. *Blood Adv* 4, 830-844, doi:10.1182/bloodadvances.2019000779 (2020).
- 2 Li, H. *et al.* Dysfunctional CD8 T Cells Form a Proliferative, Dynamically Regulated Compartment within Human Melanoma. *Cell* 176, 775-789 e718, doi:10.1016/j.cell.2018.11.043 (2019).
- 3 Matlung, H. L., Szilagyi, K., Barclay, N. A. & van den Berg, T. K. The CD47-SIRPalpha signaling axis as an innate immune checkpoint in cancer. *Immunol Rev* 276, 145-164, doi:10.1111/imr.12527 (2017).
- 4 Cai, G. *et al.* CD160 inhibits activation of human CD4+ T cells through interaction with herpesvirus entry mediator. *Nat Immunol* 9, 176-185, doi:10.1038/ni1554 (2008).
- 5 Muscate, F. *et al.* HVEM and CD160: Regulators of Immunopathology During Malaria Blood-Stage. *Front Immunol* 9, 2611, doi:10.3389/fimmu.2018.02611 (2018).
- 6 Welte, S., Kuttruff, S., Waldhauer, I. & Steinle, A. Mutual activation of natural killer cells and monocytes mediated by NKp80-AICL interaction. *Nat Immunol* 7, 1334-1342, doi:10.1038/ni1402 (2006).
- 7 Zavidij, O. *et al.* Single-cell RNA sequencing reveals compromised immune microenvironment in precursor stages of multiple myeloma. *Nature Cancer* 1, 493-506, doi:10.1038/s43018-020-0053-3 (2020).
- 8 Ledergor, G. *et al.* Single cell dissection of plasma cell heterogeneity in symptomatic and asymptomatic myeloma. *Nat Med* 24, 1867-1876, doi:10.1038/s41591-018-0269-2 (2018).

REVIEWERS' COMMENTS

Reviewer #2 (Remarks to the Author):

In this revision the authors have performed additional experiments and provide new analyses, which greatly improve the quality of the manuscript and address most of my concerns. The authors have performed experiments using flow-cytometry to validate some of their findings on a protein level. The authors have also revised some hypotheses from the original version of the manuscript, which were not sustained upon re-analysis of the data. The inherent weakness of the study is the limited validation of cell-cell interactions on a functional level. Interaction analysis based on transcriptional analysis of surface molecules in single cells by inference is inherently limited and remains hypothesis-generating. However, as such experiments can be extensive, I appreciate that deeper exploration is beyond the scope of the manuscript. The work presented in this revised version of the manuscript is of great value to the community and provides important insight into the heterogeneity of RRMM in patients with single-cell resolution and therefore contributes substantially to our understanding of the disease after treatment fails.

Reviewer #3 (Remarks to the Author):

The authors have addressed my comments satisfactorily.